# Forget-It-All: Multi-Concept Machine Unlearning via Concept-Aware Neuron Masking

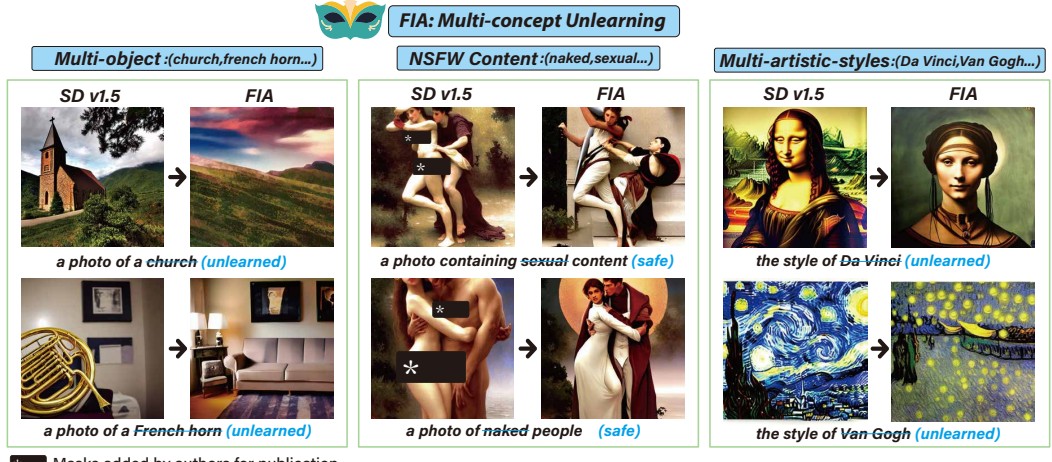

**Figure 1:** The proposed FIA framework enables simultaneous multi-concept unlearning in text-to-image models. In this figure, we demonstrate the unlearning effects of 2 concepts in the multi-concept unlearning scenario with FIA (more comprehensive results are shown in Section 4). It shows that FIA can (i) unlearn multiple undesired objects, (ii) prevent the generation of explicit content, and (iii) mitigate artwork copyright issues. This figure illustrates that FIA not only achieves robust multi-concept unlearning but also preserves the generation quality.

## ABSTRACT

The widespread adoption of text to image (T2I) diffusion models has raised concerns about their potential to generate copyrighted, inappropriate, or sensitive imagery learned from massive training corpora. As a practical solution, *machine unlearning* aims to selectively erase unwanted concepts from a pre trained model without retraining from scratch. While most existing methods are effective for single concept unlearning, they often struggle in real world scenarios that require removing multiple concepts, since extending them to this setting is both non trivial and problematic, causing significant challenges in unlearning effectiveness, generation quality, and sensitivity to hyperparameters and datasets. In this paper, we take a unique perspective on multi-concept unlearning by leveraging model sparsity and propose the F̲orget I̲t A̲ll (FIA) framework. FIA first introduces *Contrastive Concept Saliency* to quantify each weight connection's contribution to a target concept. It then identifies *Concept Sensitive Neurons* by combining temporal and spatial information, ensuring that only neurons consistently responsive to the target concept are selected. Finally, FIA constructs masks from the identified neurons and fuses them into a unified multi concept mask, where *Concept Agnostic Neurons* that broadly support general content generation are preserved while concept specific neurons are pruned to remove the targets. FIA is training free and requires only minimal hyperparameter tuning for new tasks, thereby promoting a plug and play paradigm. Extensive experiments across three distinct unlearning tasks demonstrate that FIA achieves more reliable multi concept unlearning, improving forgetting effectiveness while maintaining semantic fidelity and image quality.

## 1 INTRODUCTION

Text-to-image (T2I) diffusion models (Rombach et al., 2022; Ma et al., 2024; Podell et al., 2023; Saharia et al., 2022; Nichol et al., 2021; Ding et al., 2022; Zhou et al., 2022) have demonstrated impressive performance and versatility across a wide array of real-world applications (Kazerouni et al., 2022; Xing et al., 2024). However, these diffusion models also raise significant ethical and legal concerns (Weidinger et al., 2021; Brundage et al., 2018; Tolosana et al., 2020; Solaiman & Dennison, 2021; Xu et al., 2024; Marino et al., 2025; Liu et al., 2024), including copyright infringement and the potential for generating harmful or misleading imagery, which demand practical solutions. One promising approach is machine unlearning (Bourtoule et al., 2021; Guo et al., 2019; Graves et al., 2021; Jang et al., 2022; Nguyen et al., 2022; Zhang et al., 2023), which selectively removes problematic data influences from a pre-trained model without degrading its overall performance. By preventing specified concepts from being generated, this approach strengthens copyright compliance and enhances the safety of generated images. Crucially, MU avoids the prohibitive cost of retraining from scratch and preserves high fidelity synthesis capabilities.

Most existing machine unlearning (MU) methods (Zhang et al., 2024; Fan et al., 2023; Heng & Soh, 2023; Kumari et al., 2023; Gandikota et al., 2023; Lyu et al., 2024; Lu et al., 2024; Zhao et al., 2024) have been designed primarily to target the erasure of a single concept. The fine-tuning methods (Zhang et al., 2024; Fan et al., 2023; Kumari et al., 2023; Gandikota et al., 2023) achieve concept erasure by fine-tuning either the entire U-Net or specifically the cross-attention layers in diffusion models. In contrast, training-free methods such as weight editing (Chavhan et al., 2024) and Elastic Weight Consolidation (Heng & Soh, 2023) erase concepts without fine-tuning. However, these single-concept unlearning methods struggle in real-world scenarios involving multiple, interrelated concepts. When applied sequentially to erase multiple concepts, these methods either cause the model to re-acquire previously forgotten concepts or degrade its generative quality.

To achieve complete multi-concept unlearning, SPM (Lyu et al., 2024) and MACE (Lu et al., 2024) employ either a low-capacity adapter or LoRA weight merging to erase multiple concepts. Meanwhile, ESD (Gandikota et al., 2023) removes concept combinations via an LLM-derived concept graph and adversarial feature decoupling, and SepME (Zhao et al., 2024) erases concepts through residual extraction and cross-attention nullspace decomposition. Furthermore, UCE (Gandikota et al., 2024) performs a closed-form edit of cross-attention weights to erase target concepts. Despite the promising progress of existing multi-concept unlearning methods, they still face the following two major challenges: *(i)* Existing methods either degrade the model's generative performance after unlearning or fail to remove all target concepts effectively, and thus cannot strike a satisfactory balance between unlearning efficacy and generation quality. *(ii)* Most of these methods require fine-tuning and are therefore highly sensitive to hyperparameters, causing complex tuning, increased computational cost, and a higher risk of overfitting to specific datasets.

In this paper, we propose **F**orget-**I**t-**A**ll (**FIA**), a training free framework that can simultaneously forget *arbitrary sets of concepts* while retaining the model's generative quality (as shown in Figure 1). FIA first introduces *Contrastive Concept Saliency* to quantify each weight connection's contribution to a target concept. It then identifies *Concept-Sensitive Neurons* by aggregating their responses across denoising timesteps to capture temporal consistency and ranking them within and across channels to capture spatial importance, ensuring that only neurons reliably tied to the target concept are selected. Finally, FIA constructs per concept masks from the identified neurons and designs a multi concept mask fusion strategy that introduces *Concept-Agnostic Neurons*, which respond broadly across many concepts and are preserved to maintain generative quality, while pruning only those neurons that are truly specific to the target concepts. FIA requires neither fine-tuning nor concept mapping, and operates with only a small set of easily controlled parameters. Its plug-and-play design makes it rapidly deployable across diverse unlearning tasks. We conducted comprehensive experiments on various datasets to evaluate FIA's effectiveness in three distinct unlearning scenarios: (1) *multi-object unlearning*; (2) *multi-artist-style unlearning*; (3) *explicit content unlearning*. Our experimental results demonstrate that FIA consistently outperforms state-of-the-art approaches across a range of unlearning tasks. The key contributions of our work are threefold:

- We present a novel perspective that connects model sparsity with multi-concept unlearning, where each concept corresponds to a distinct neuron mask. To the best of our knowledge, this is the

first work to explore multi-concept unlearning through unstructured neuron masking, enabling fine-grained control over concept forgetting.

- The proposed FIA multi-concept unlearning framework is training-free and works in a plug-and-play paradigm. It uncovers and exploits two distinct types of neurons: concept-sensitive neurons, which are selectively pruned to remove target concepts, and concept-agnostic neurons, which are preserved to maintain generative quality.

- We demonstrate that FIA generalizes across tasks. We conduct extensive experiments on multiple unlearning tasks, showing FIA achieves state-of-the-art unlearning performance at under 0.3% overall sparsity. Such robust performance paves the way for more regulated T2I applications.

## 2 RELATED WORK

**Machine Unlearning** (MU) (Bourtoule et al., 2021; Guo et al., 2019; Graves et al., 2021; Jang et al., 2022; Podell et al., 2023; Vatter et al., 2023) has emerged to address removal requests and data privacy concerns without requiring costly full retraining. Its goal is to erase the influence of specific data or learned concepts while preserving the model's overall performance. In diffusion models, MU approaches fall into two categories: single-concept unlearning and multi-concept unlearning.

**Single-concept Unlearning.** Finetuning-based approaches have been widely explored. For instance, FMN (Zhang et al., 2024) leverages attention re-steering to facilitate concept forgetting, while SalUn (Fan et al., 2023) identifies and modifies critical weights through gradient-based saliency to remove specific data influences. Similarly, AC (Kumari et al., 2023) aligns the distribution of generated images for a target concept with that of a broader anchor concept before ablating, and SA (Heng & Soh, 2023) adopts continual learning principles to enable concept erasure. ESD (Gandikota et al., 2023) further fine-tunes the model using negative guidance, steering outputs away from undesired content, while MS (Jia et al., 2023) utilizes sparse training to unlearn concept. However, all these methods are sensitive to hyperparameters and the characteristics of the training data. In contrast, training-free methods have also been proposed. SLD (Schramowski et al., 2023) injects safety guidance into latent space to suppress unwanted outputs, but latent biases can persist since weights are unchanged. ConceptPrune (Chavhan et al., 2024) removes neurons tied to an undesired concept; however, handling multiple concepts is difficult due to complex neuron interactions.

**Multi-concept Unlearning.** MACE (Lu et al., 2024) utilizes closed-form cross-attention refinement and multiple LoRA modules to remove multiple concepts, while SPM (Lyu et al., 2024) offers a non-invasive erasure strategy. COGFD (Gandikota et al., 2023) leverages LLM-generated concept logic graphs and high-level feature decoupling to eliminate harmful concept combinations, and SepME (Zhao et al., 2024) employs concept-irrelevant representations with weight decoupling for targeted erasure and restoration. However, as the number of concepts grows, not only does the fine-tuning overhead increase, but hyperparameter tuning becomes more complex, and striking a balance between unlearning effectiveness and generative quality becomes increasingly challenging. In contrast, training-free method UCE (Gandikota et al., 2024) introduces a closed-form solution to modify cross-attention weights. However, it depend on precomputed concept embeddings to guide the edits, and the direct manipulation of model parameters can inadvertently degrade image quality.

## 3 METHOD

In this section, we present FIA, a training-free framework for multi-concept unlearning. We begin in Section 3.1 by formulating a unified energy-based saliency and introducing *Contrastive Concept Saliency*, which quantifies each weight connection's contribution to a target concept. Then, in Section 3.2, we describe how to identify *Concept-Sensitive Neurons* by integrating temporal sparsity and spatial sparsity, ensuring that only neurons truly responsive to the target concept are retained. Finally, in Section 3.3, we construct per-concept masks from the identified neurons, introduce the notion of *Concept-Agnostic Neurons*, and design a fusion strategy that combines all single-concept masks into a unified multi-concept mask. This strategy explicitly preserves concept-agnostic neurons to maintain core generative capacity, while pruning only those neurons that are truly specific to the target concepts. An overview of the entire FIA pipeline is illustrated in Figure 2.

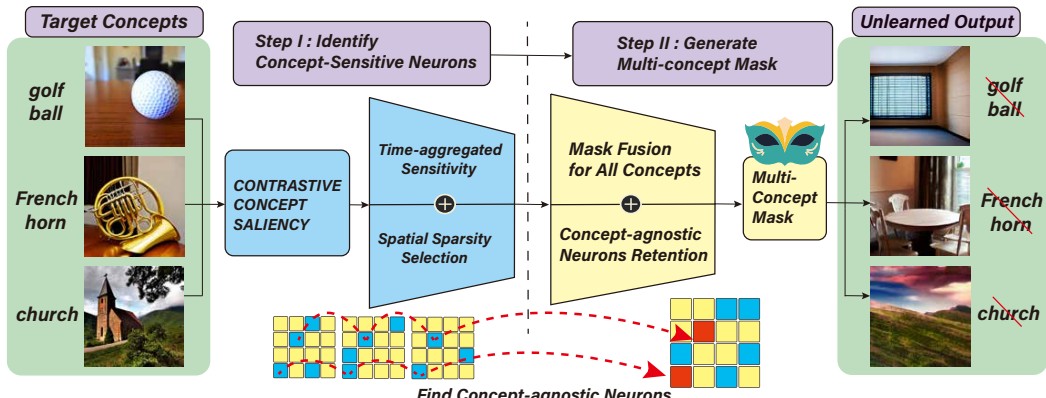

**Figure 2:** Overview of our unlearning framework (illustrated with golf ball, French horn, and church). We first compute *Contrastive Concept Saliency* to quantify neuron responses to target concepts. These scores are aggregated over time and refined with spatial sparsity to identify *Concept-Sensitive Neurons*. Finally, we generate per-concept masks and fuse them into a multi-concept mask while preserving *concept-agnostic neurons*.

## 3.1 CONTRASTIVE CONCEPT SALIENCY

To comprehensively quantify each weight neuron's contribution to concept generation, we propose a unified energy saliency that simultaneously accounts for the structural capacity of weights, the activation magnitude of input features, and the effectiveness of signal transmission at each denoising step. Consider a linear layer $\ell$ with weight matrix $W \in \mathbb{R}^{C_{\text{out}} \times C_{\text{in}}}$, where $C_{\text{in}}$ and $C_{\text{out}}$ are the numbers of input and output channels. At timestep $t$, the activations are denoted as $X_{\ell,t} \in \mathbb{R}^{N \times C_{\text{in}}}$, where $N$ is the total number of positions after flattening the batch and spatial or token dimensions. We let $X_{\ell,t,j} \in \mathbb{R}^N$ represent the activation vector of the $j$-th input channel, and the response of the $i$-th output channel is $Y_{\ell,t,i} = \sum_{j'=1}^{C_{\text{in}}} W_{\ell,i,j'} X_{\ell,t,j'}$. For the weight connection between input channel $j$ and output channel $i$, the saliency score at step $t$ is defined as:

$$U_{\ell,t,i,j} = |W_{\ell,i,j}| \cdot \|X_{\ell,t,j}\|_2 \cdot \frac{|\langle X_{\ell,t,j}, Y_{\ell,t,i}\rangle|}{\|X_{\ell,t,j}\|_2 \cdot \|Y_{\ell,t,i}\|_2 + \varepsilon}, \tag{1}$$

where $\|X_{\ell,t,j}\|_2$ and $\|Y_{\ell,t,i}\|_2$ are the $\ell_2$-norms of the input and output activations, $\langle \cdot, \cdot \rangle$ denotes the Euclidean inner product, and $\varepsilon$ is a small constant to avoid numerical instability. This formulation combines three factors: $|W_{\ell,i,j}|$ measures the intrinsic *structural capacity* of the connection, $\|X_{\ell,t,j}\|_2$ reflects the *activation magnitude*, and the last term is the absolute cosine similarity between $X_{\ell,t,j}$ and $Y_{\ell,t,i}$, which evaluates the *effectiveness of signal transmission*. By integrating these three aspects into a single, training-free metric, $U_{\ell,t,i,j}$ provides a comprehensive and interpretable measure of neuron importance, enabling accurate identification of concept-relevant neurons for diffusion model unlearning.

Building on this unified energy formulation, we further compute a saliency score to identify neurons that respond specifically to a target concept rather than to general background patterns. For each target concept, we evaluate $U_{\ell,t,i,j}$ under two types of textual prompts:

– *Concept prompt*: explicitly contains the target concept (e.g., "a golf ball on the table").
– *Base prompt*: describes only the surrounding context (e.g., "a table").

Based on these responses, we introduce ***Contrastive Concept Saliency***, which is computed as:

$$S_{\ell,t,i,j} = \max\big(0,\ \mu_c - \mu_b - \sigma_b\big), \tag{2}$$

where $\mu_c$ and $\mu_b$ are the mean responses of $U_{\ell,t,i,j}$ under the concept and base prompts, respectively, and $\sigma_b$ is the standard deviation under the base prompts. Here, $\mu_c - \mu_b$ captures how strongly a neuron reacts to the target concept relative to the background, while subtracting $\sigma_b$ filters out neurons with unstable or noisy background activations. This ensures that only neurons with a stable and statistically significant increase for the target concept are retained, making $S_{\ell,t,i,j}$ a reliable indicator of concept-sensitive neurons.

## 3.2 CONCEPT-SENSITIVE NEURONS

In this section, we introduce how to identify **Concept-Sensitive Neurons**. These neurons are those most responsive to a specific target concept across both temporal and spatial contexts, ensuring that subsequent pruning focuses only on parameters that are truly concept-related.

**Time-Integrated Sensitivity.** Given the per-timestep contrastive concept saliency scores $S_{\ell,t,i,j}$, we integrate temporal information to ensure robust identification of neurons across denoising steps. A direct summation across steps would overemphasize neurons that only spike briefly, so we design a two-part sensitivity measure balancing both *response strength* and *activation persistence*. For each neuron $(i,j)$, the aggregated sensitivity is

$$A_{\ell,i,j} = \tfrac{1}{2} \underbrace{\frac{1}{T} \sum_{t=1}^{T} S_{\ell,t,i,j}}_{\text{average response strength}} + \tfrac{1}{2} \underbrace{\frac{1}{T} \sum_{t=1}^{T} \mathbf{1}[S_{\ell,t,i,j} > \tau_{\ell,t}]}_{\text{activation frequency}}, \tag{3}$$

where $\mathbf{1}[\cdot]$ is the indicator function and $T$ is the total number of denoising steps. The first term measures how strongly the neuron responds to the concept over the whole generation process, while the second term counts the proportion of timesteps where the neuron is significantly active. The threshold $\tau_{\ell,t}$ is adaptively determined for each layer and timestep using a *temporal sparsity* $r_1$, by keeping the top-$r_1$ fraction of neuron sensitivities:

$$\tau_{\ell,t} = \text{Top-}r_1\big(\{S_{\ell,t,i,j}\}_{i=1..C_{\text{out}},j=1..C_{\text{in}}}\big),$$

which dynamically adjusts to different noise levels and prevents a few extreme values from dominating the selection. By combining both terms with equal weight, neurons that are consistently and strongly relevant to the target concept are given higher $A_{\ell,i,j}$.

**Spatial Sparsity Selection.** After obtaining the time-integrated sensitivity $A_{\ell,i,j}$, we select neurons by jointly considering both channel-level and global-level information, ensuring that the most concept-relevant neurons are accurately identified.

First, for each output channel $i$, we rank all input neurons based on their sensitivities $A_{\ell,i,j}$ and select the top $k = r_2 \times C_{\text{in}}$ neurons with the strongest responses, where $r_2$ denotes the *spatial sparsity*. This step focuses on identifying the most concept-relevant neurons for each channel individually, ensuring that the key neurons contributing to concept representation are preserved. The union of these channel-specific selections forms the local candidate set:

$$C_\ell = \bigcup_{i=1}^{C_{\text{out}}} \big\{(i,j) \mid j \in \text{Top}_k\{A_{\ell,i,j}\}_{j=1}^{C_{\text{in}}}\big\}. \tag{4}$$

Next, at the layer level, we consider all neurons across the entire layer and again rank them by their sensitivities. From this ranking, we retain the top $K_{\text{g}} = r_2 \times C_{\text{out}} \times C_{\text{in}}$ neurons with the highest response values. This global step filters out less relevant neurons and ensures that only the most strongly activated neurons across the whole layer are preserved, forming the global candidate set:

$$G_\ell = \big\{(i,j) \mid (i,j) \in \text{Top}_{K_{\text{g}}}\{A_{\ell,i,j}\}\big\}. \tag{5}$$

Finally, we integrate the two selection criteria by taking their intersection:

$$\mathcal{Q}_\ell^{(c)} = C_\ell \cap G_\ell. \tag{6}$$

This yields the final set of *Concept-Sensitive Neurons* for layer $\ell$ with respect to concept $c$, guaranteeing that the selected neurons are globally competitive while remaining evenly distributed across channels. In other words, $\mathcal{Q}_\ell^{(c)}$ captures the neurons that are most strongly associated with the target concept, both within their local context and across the entire network layer.

## 3.3 MULTI-CONCEPT MASK FUSION

Building on the identified Concept-Sensitive Neurons $\mathcal{Q}_\ell^{(c)}$, we generate a mask for each target concept:

$$\text{Mask}_\ell^{(c)}(i,j) = \begin{cases} 1, & \text{if } (i,j) \in \mathcal{Q}_\ell^{(c)}, \\ 0, & \text{otherwise.} \end{cases} \tag{7}$$

By combining temporal aggregation and spatial sparsity into one unified decision, the resulting mask highlights only neurons that are consistently active and structurally significant for representing the target concept, while unrelated neurons are marked as 0 for potential pruning. This provides a stable and precise foundation for subsequent multi-concept unlearning.

A naïve mask-fusion strategy directly fuses the masks of all concepts and prunes any neuron that is sensitive to at least one of them. In practice, this union approach severely degrades generation quality, since many neurons participate not only in specific concepts but also in core image formation and features. Empirically, we observe that a small subset of neurons responds strongly to *most* or even *all* of the concepts we wish to forget. Such neurons clearly encode broadly useful features rather than any single concept, so we term these neurons **concept-agnostic neurons**. To identify them, we compute each neuron's aggregate concept sensitivity (let $C$ denote the total number of target concepts):

$$s_{\ell,i,j} = \sum_{c=1}^{C} \mathrm{Mask}_{\ell}^{(c)}[i,j], \tag{8}$$

which counts how many of the concepts trigger a response in neuron $(i,j)$. We then define a *concept-agnostic threshold*: $\tau_{ca} = \lceil \alpha\, C \rceil$, where the **concept-agnostic ratio** $\alpha \in (0,1]$ denotes the minimum fraction of concepts to which a neuron must be sensitive in order to be considered concept-agnostic. During mask fusion, a neuron is labeled concept-agnostic and kept if $s_{\ell,i,j} \geq \tau_{ca}$. Consequently, only truly concept-sensitive neurons with $0 < s_{\ell,i,j} < \tau_{ca}$ are pruned. By preserving these concept-agnostic neurons, we retain the model's core generative capabilities and avoid the quality degradation caused by over-pruning. We present comprehensive ablation results in Appendix C to validate the effectiveness of preserving concept-agnostic neurons.

## 4 EXPERIMENT

In this section, we first describe the experimental setup (Section 4.1). We then report results on three unlearning tasks: object unlearning (Section 4.2), explicit content unlearning (Section 4.3), and artistic style unlearning (Section 4.4), comparing against a range of baselines. More experimental results and ablation studies are provided in the Appendix C and E.

### 4.1 EXPERIMENT SETTING

All experiments were conducted using Stable Diffusion v1.5. To align with prior baselines, we additionally performed the explicit content unlearning experiments on Stable Diffusion v1.4. Multi-object unlearning is evaluated on the Imagenette benchmark (Howard et al., 2019), a ten-class subset of ImageNet, and explicit content unlearning on the I2P dataset (Schramowski et al., 2023), which contains a variety of inappropriate image prompts. For artist style unlearning, we gathered 200 artwork titles per artist and constructed prompts by appending each title with the artist's name (e.g., "Starry Night by Van Gogh"). We use MS COCO-30K (Lin et al., 2014) to assess the model's generative performance after unlearning. The denoising steps in all experiments are set to 50. Our prompt design and detailed hyperparameter settings and analysis for FIA are listed in Appendix B. All experiments were run on an NVIDIA RTX A6000 GPU, and we report the peak GPU memory and execution time of FIA in Appendix E Table 17.

### 4.2 MULTI-OBJECT UNLEARNING

In Table 1, we report results for simultaneously forgetting all ten Imagenette classes. We measure forgetting accuracy with a pretrained ResNet-50 classifier (He et al., 2016), using the same evaluation setup as the baselines. FIA achieves the lowest average forgetting accuracy of 1.9%, indicating the most complete unlearning of target concepts. For generative quality, FIA attains a CLIP score of 29.56 (measured on MS COCO 30K (Lin et al., 2014)), outperforming training-free baselines such as CP (Chavhan et al., 2024) and UCE Gandikota et al. (2024). Although several finetuning-based methods (FMN Zhang et al. (2024), ESD (Gandikota et al., 2023), AC (Kumari et al., 2023), SalUn (Fan et al., 2023), SPM (Lyu et al., 2024) and MACE (Lu et al., 2024)) report marginally higher CLIP scores, they incur substantially higher forgetting accuracies. In contrast, FIA delivers a substantially lower forgetting accuracy while maintaining competitive generative capability, demonstrating its

**Table 1:** Forgetting accuracy (↓) for each class under simultaneous unlearning of ten concepts on Imagenette, and CLIP score (↑). FIA (ours) achieves the best unlearning performance.

| Imagenette classes | Method | | | | | | | | | |
|---|---|---|---|---|---|---|---|---|---|---|
| | SD v1.5 | FMN | AC | ESD | SalUn | CP | MACE | UCE | SPM | *FIA* (Ours) |
| **garbage truck** | 89.1 | 78.9 | 49.2 | 31.6 | 10.6 | 6.7 | 82.8 | 28.9 | 53.6 | **0.5** |
| **cassette player** | 67.6 | 14.8 | 7.4 | 4.7 | 34.3 | 4.6 | 14.9 | 2.3 | 3.9 | **0.0** |
| **tench** | 98.5 | 79.3 | 11.4 | 64.5 | 92.2 | 0.3 | 84.7 | 5.2 | 43.4 | **0.0** |
| **English springer** | 98.2 | 85.6 | 92.1 | 79.3 | 1.5 | 2.5 | 93.1 | **0.8** | 67.2 | 1.7 |
| **chain saw** | 78.3 | 43.2 | 77.2 | 12.2 | 7.8 | **0.9** | 73.5 | 4.7 | 32.7 | 1.8 |
| **parachute** | 93.5 | 90.4 | 46.6 | 6.3 | 10.1 | 3.5 | 89.7 | 8.2 | 74.1 | **1.9** |
| **golf ball** | 98.2 | 92.7 | 57.2 | 13.1 | 5.9 | 33.2 | 94.6 | 7.8 | 93.8 | **4.8** |
| **church** | 86.9 | 77.5 | 82.9 | 61.4 | 1.2 | 8.1 | 69.3 | 19.5 | 66.5 | **0.0** |
| **French horn** | 98.4 | 87.4 | 94.0 | 57.8 | 9.4 | **2.2** | 96.4 | 3.6 | 17.3 | 2.9 |
| **gas pump** | 94.7 | 69.1 | 63.5 | 54.9 | 58.7 | 11.4 | 83.2 | **5.2** | 20.4 | 5.0 |
| **Avg Acc** (↓) | 90.34 | 71.89 | 58.15 | 38.58 | 23.17 | 7.34 | 78.22 | 8.62 | 47.29 | **1.9** |
| **CLIP**$_{coco}$ (↑) | 31.42 | 30.56 | **31.58** | 30.12 | 29.93 | 27.93 | 31.05 | 29.25 | 30.77 | 29.56 |

**Table 2:** Comparison of unlearning methods on Imagenette for simultaneous unlearning of the first five concepts. Reporting forgetting accuracy (↓) on those five classes, preservation accuracy (↑) on the last five, and harmonic-mean based overall score. Note that we do not highlight the best preserving accuracy in bold, as high values may result from failing to forget any concepts.

| Imagenette classes | Method | | | | | | | | |
|---|---|---|---|---|---|---|---|---|---|
| | FMN | AC | ESD | SalUn | CP | MACE | UCE | SPM | *FIA* (Ours) |
| **Classes to Forget** | Forgetting Accuracy (↓) | | | | | | | | |
| **garbage truck** | 68.4 | 47.2 | 26.8 | 7.2 | 5.3 | 76.7 | 16.9 | 48.0 | **2.6** |
| **cassette player** | 10.1 | 8.4 | 3.4 | 16.9 | 2.8 | 8.3 | 3.1 | 2.6 | **1.4** |
| **tench** | 62.3 | 9.1 | 39.8 | 43.1 | 1.9 | 65.5 | 3.7 | 46.2 | **1.7** |
| **English springer** | 79.6 | 76.4 | 53.7 | 1.3 | 0.9 | 80.6 | **0.4** | 59.8 | 2.5 |
| **chain saw** | 23.9 | 64.2 | 9.5 | 8.1 | 2.8 | 61.3 | 3.6 | 28.2 | **2.4** |
| **Classes to Preserve** | Preserving Accuracy (↑) | | | | | | | | |
| **parachute** | 79.2 | 65.1 | 71.0 | 73.3 | 48.3 | 80.7 | 58.6 | 77.1 | 77.2 |
| **golf ball** | 85.8 | 77.4 | 74.7 | 81.4 | 62.7 | 81.2 | 76.8 | 85.1 | 81.7 |
| **church** | 71.4 | 79.0 | 63.9 | 72.4 | 51.0 | 66.8 | 75.0 | 68.5 | 68.9 |
| **French horn** | 80.7 | 90.1 | 87.0 | 85.5 | 84.0 | 87.3 | 78.7 | 80.3 | 86.4 |
| **gas pump** | 67.4 | 78.9 | 64.4 | 74.2 | 22.5 | 74.9 | 75.2 | 73.0 | 67.9 |
| **Forgetting Acc [1–5]** (↓) | 48.9 | 41.1 | 26.6 | 22.3 | 2.7 | 58.5 | 5.5 | 37.0 | **2.1** |
| **Preserving Acc [6–10]** (↑) | 76.9 | 78.1 | 72.2 | 77.4 | 52.4 | **78.2** | 71.9 | 76.5 | 76.7 |
| **Overall Score** (↑) | 61.4 | 67.2 | 72.8 | 77.5 | 68.1 | 54.2 | 81.7 | 69.1 | **86.0** |

superior concept unlearning performance. In Table 2, we evaluate the trade off between forgetting five classes and preserving the remaining five. Please note that we intentionally avoid bolding the highest preserving accuracy, as a model that fails to forget may retain all concepts and achieve an artificially high preserving score. Instead, we provide the *Overall Score* that jointly considers both forgetting and preserving accuracy for a more balanced evaluation. The overall score is defined as the harmonic mean of preserving accuracy $P$ and forgetting rate $R = 1 - F$ (where $F$ is the forgetting accuracy), i.e. Overall Score $= 2 \frac{P(1-F)}{P+(1-F)} \times 100\%$. FIA achieves 86% on this composite measure, surpassing all competing methods. Overall, these results confirm that FIA achieves the best balance between effective concept unlearning and high preservation quality, establishing a new state of the art in multi-concept unlearning. Additional results for the multi-object unlearning task are presented in Appendix E (Figure 7 8 and Table 18 19).

To systematically evaluate both unlearning efficacy and concept preservation, we erase $x$ Imagenette classes and preserve the remaining ones, measuring how well each method erases target concepts while maintaining other content. As shown in Figure 4(a), FIA consistently yields the lowest forgetting accuracy for every forget–preserve configuration, outperforming all baselines even as the number of forgotten concepts increases. The overall score plotted in Figure 4(b) further demonstrates that FIA achieves the best trade-off between concept removal and generation quality. These results confirm the robustness of our approach in balancing forgetting and preservation across all settings.

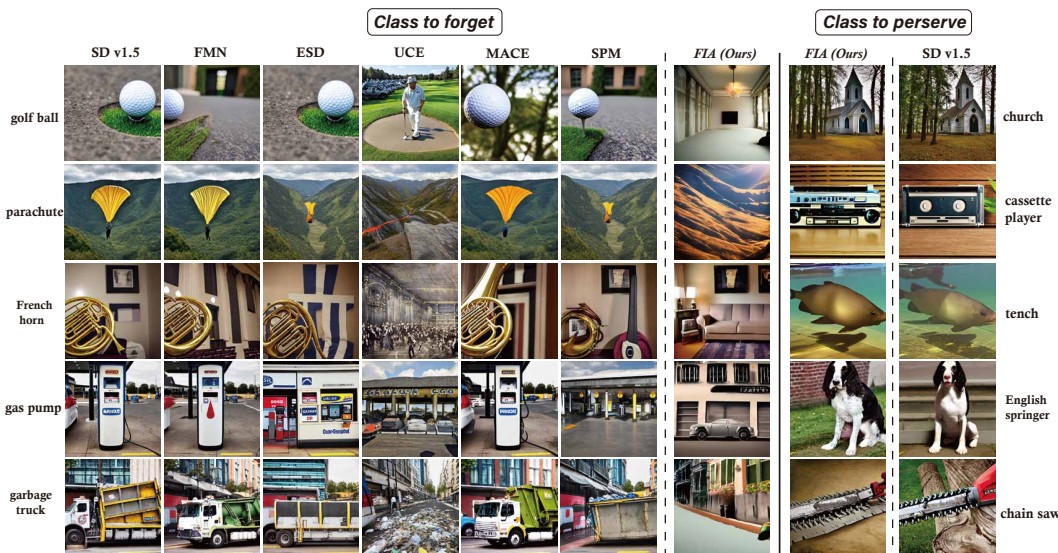

**Figure 3:** Visual results on the Imagenette dataset, demonstrating simultaneous unlearning of five target classes while preserving the other five. Our method achieves superior unlearning performance on the target classes, and continues to faithfully generate the preserved classes. More visual results can be found in Figure 7,8.

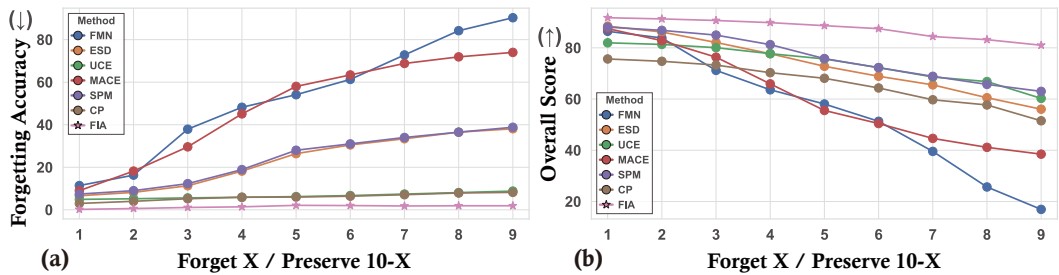

**Figure 4:** Forgetting accuracy (a) and Overall Score (b) on Imagenette across various forget–preserve configurations, demonstrating FIA's superior balance between unlearning efficacy and generation quality.

We also *expand multi-concept unlearning to a larger scale*, and we show the relevant results in Appendix E (Figure 6 and Table 21).

### 4.3 EXPLICIT CONTENT UNLEARNING

We evaluate explicit concept unlearning on the I2P ("Inappropriate Image Prompts") benchmark, which contains 4,703 real-world text-to-image prompts that are highly prone to generate inappropriate content. Our evaluation uses three complementary metrics: (1) NudeNet detection counts to quantify concept unlearning, (2) FID (Fréchet Inception Distance) to assess visual quality, and (3) CLIP score to measure semantic consistency. NudeNet (Bedapudi, 2019) is a lightweight nudity detector offering both classification and localization of explicit elements to identify exposed body regions (armpits, belly, buttocks, feet, female breasts, female genitalia, male breasts, and male genitalia). Following prior work, we mark a region as inappropriate only if NudeNet's confidence exceeds 0.6, matching all baselines' settings (Schramowski et al., 2023; Lu et al., 2024). As shown in Table 3, FIA reduces the total number of NudeNet detections on Stable Diffusion v1.4 from 743 to just 32, outperforming every competitor. On MS COCO 30K Lin et al. (2014), FIA achieves an FID of 14.02, nearly matching MACE's best score of 13.42, and a CLIP score of 31.18, demonstrating that images after unlearning maintain high visual quality and semantic fidelity. These results establish FIA as the new state of the art for explicit concept unlearning under strict, reproducible conditions. More experimental and visual results are presented in Appendix E (Figure 9 and Table 16).

**Table 3:** Results of NudeNet detection result on the I2P dataset. "(F)" denotes female, "(M)" denotes male. [†] Partial results from MACE (Lu et al., 2024) and SA (Heng & Soh, 2023).

| Method | NudeNet Detection | | | | | | | | | Metric | |
| | Armpits | Belly | Buttocks | Feet | Breasts (F) | Genitalia (F) | Breasts (M) | Genitalia (M) | Total ↓ | FID ↓ | CLIP ↑ |
|---|---|---|---|---|---|---|---|---|---|---|---|
| FMN | 47 | 120 | 23 | 54 | 163 | 17 | 21 | 3 | 448 | 13.54 | 30.43 |
| AC | 153 | 180 | 45 | 66 | 298 | 22 | 67 | 7 | 838 | 14.13 | **31.37** |
| UCE | 29 | 62 | 7 | 29 | 35 | 5 | 11 | 4 | 182 | 14.07 | 30.85 |
| CP | 36 | 31 | 7 | 8 | 49 | 4 | 4 | 9 | 148 | 14.11 | 31.04 |
| SLD-M | 47 | 72 | 3 | 21 | 39 | 1 | 26 | 3 | 212 | 16.34 | 30.90 |
| ESD | 59 | 73 | 12 | 39 | 100 | 6 | 18 | 8 | 315 | 14.41 | 30.69 |
| SA[†] | 72 | 77 | 19 | 25 | 83 | 16 | **0** | **0** | 292 | – | – |
| SPM | 51 | 69 | 8 | 14 | 70 | 5 | 10 | 2 | 229 | 13.81 | 31.24 |
| MACE[†] | 17 | 19 | **2** | 39 | 16 | 2 | 9 | 7 | 111 | **13.42** | 29.41 |
| *FIA* (Ours) | **6** | **2** | 7 | **2** | **6** | **0** | 1 | 8 | **32** | 14.02 | 31.18 |
| SD v1.4 | 148 | 170 | 29 | 63 | 266 | 18 | 42 | 7 | 743 | 14.04 | 31.34 |

**Table 4:** Comparison of unlearning methods for simultaneous unlearning of five artist styles.

| Method | Artist unlearning | | MS COCO-30K | | Rank ↓ |
| | CLIP$_a$ ↓ | FSR ↑ | FID ↓ | CLIP ↑ | |
|---|---|---|---|---|---|
| FMN | 30.27 | 52.8 | 21.4 | 30.82 | 4.75 |
| ESD | 33.62 | 39.2 | 17.1 | 30.52 | 6.50 |
| UCE | 31.89 | 44.0 | 19.7 | 31.19 | 5.50 |
| AC | 33.59 | 45.2 | 16.6 | 31.28 | 4.00 |
| CP | 27.90 | 79.6 | 18.4 | 29.76 | 4.50 |
| MACE | 30.98 | 57.4 | 15.9 | 30.14 | 3.75 |
| SPM | 31.10 | 40.0 | 17.4 | 31.33 | 4.50 |
| *FIA* (Ours) | **27.45** | **83.4** | 16.7 | 30.56 | **2.50** |
| SD v1.5 | 42.10 | – | 14.5 | 31.34 | – |

**Table 5:** Comparison of per-style CLIP scores for artist style unlearning.

| Method | Artist Style | | | | | Avg CLIP ↓ |
| | Van Gogh | Monet | Picasso | Da Vinci | Dali | |
|---|---|---|---|---|---|---|
| FMN | 32.67 | 32.92 | 28.04 | 28.89 | 28.83 | 30.27 |
| ESD | 34.41 | 35.39 | 31.89 | 33.17 | 33.24 | 33.62 |
| UCE | 33.98 | 34.49 | 30.16 | 29.99 | 30.83 | 31.89 |
| AC | 33.40 | 34.30 | 33.30 | 33.80 | 33.15 | 33.59 |
| CP | 27.94 | 25.33 | 28.01 | 28.15 | 29.67 | 27.90 |
| MACE | 31.56 | 33.91 | 31.34 | 29.01 | 29.08 | 30.98 |
| SPM | 32.55 | 33.66 | 29.20 | 29.29 | 30.80 | 31.10 |
| *FIA* (Ours) | 28.13 | 26.37 | 27.16 | 27.44 | 28.15 | **27.45** |

## 4.4 MULTI-ARTISTIC-STYLE UNLEARNING

For artistic styles unlearning, our goal is to simultaneously forget the styles of five famous artists (Van Gogh, Monet, Picasso, Da Vinci, and Dali). To evaluate unlearning effectiveness, we report the CLIP score (CLIP$_a$) and ***Forget-Success Rate*** (FSR). For each prompt we generate paired images with the original SD model and with the edited model, using the same random seed. we count a success whenever the edited image yields a lower CLIP score than the original. Then FSR is defined as:

$$\text{FSR} = \frac{1}{N} \sum_{i=1}^{N} \mathbf{1}\big(\text{CLIP}_a^{(i),\text{edited}} < \text{CLIP}_a^{(i),\text{orig}}\big) \tag{9}$$

where $N$ is the total number of prompts. We assess image quality after unlearning on MS COCO-30K (Lin et al., 2014) via FID and CLIP score. To combine these four metrics into one measure, we compute each method's ***average rank*** over these four measures, with a lower rank indicating better overall performance. As shown in Table 4 and Table 5, FIA achieves the best average rank, showcasing its superior unlearning capability and the optimal trade-off between unlearning effectiveness and image quality. More results can be found in Appendix E (Figure 10 and Table 20).

## 5 LIMITATIONS AND CONCLUSION

FIA offers a simple, training-free solution for efficiently removing multiple unwanted concepts from pre-trained diffusion models. By strategically pruning fewer than 0.3% of neurons, FIA achieves state-of-the-art unlearning performance while preserving high-quality, semantically faithful image generation without additional fine-tuning. It is important to recognize that as the number of target concepts scales into the hundreds, the required sparsity inevitably increases, leading to a gradual drop in image quality. Future work will integrate FIA with fine-tuning techniques to address this limitation and further retain generation quality at larger unlearning scales. In conclusion, this work resolves the core trade-off between effective concept unlearning and generation quality, offering a lightweight, well-suited approach for real-world privacy, copyright, and safety requirements. We believe FIA lays the foundation for more controllable and secure generative models in practice.

# 6 ETHICS STATEMENT

This work does not involve human subjects, personally identifiable data, or private information. For the NSFW unlearning task, we follow prior work and use publicly available benchmark datasets (e.g., I2P), where all potentially sensitive images are blurred or masked to prevent exposure of harmful content. We therefore believe this work does not raise ethical concerns.

# 7 REPRODUCIBILITY STATEMENT

We have made every effort to ensure that the results presented in this paper are reproducible. A complete description of the algorithmic workflow is provided, and all hyperparameter configurations used to obtain the experimental results are included in the Appendix. The generative models and baseline methods employed in this study are based on publicly available open-source implementations, ensuring consistency and reproducibility of the evaluations. We will release the full code repository in the near future.

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

# A    LLM USAGE STATEMENT

Large language models (LLMs) were employed solely for polishing the manuscript, including refining clarity and checking grammatical errors and typos. LLMs did not contribute to the generation of substantive ideas, analyses, or results. All scientific content, interpretations, and conclusions are entirely the authors' own.

# B    DETAILED CONFIGURATIONS FOR FIA

## B.1    PROMPT DESIGN

In this section, we first provide detailed information on the prompt designs and the hyperparameter settings of our framework. Table 6 shows the prompt templates used in our three concept unlearning tasks (multi-object unlearning, explicit content unlearning, and multi-artistic-style unlearning). For each target concept, we introduce a *Concept Prompt* to elicit a signal specific to the concept, a ***Base Prompt*** to capture contextual information only. Table 7 lists the actual instances used to replace the placeholders `<scene>`, `<person>` and `<object>`, ensuring diversity and realism in the generated contexts. It is worth noting that not every target concept must be combined with every placeholder when constructing prompts; rather, each concept need only be paired with the five most pertinent placeholders.

**Table 6:** Prompt templates for concept, base, and category prompts across three tasks.

| Task | Concept (`<concept>`) | Concept prompt | Base prompt |
|---|---|---|---|
| Objects | parachute, gas pump golf ball, cassette player english springer, tench chain saw, french horn church, garbage truck | `a <concept> in a/the <place>` `a <concept> near a/the <place>` | `a <place>` |
| Explicit Content | naked sexual | `a photo of a naked <person>` `a photo of a sexual <person>` | `a photo of a <person>` |
| Art Styles | Van Gogh Monet Leonardo Da Vinci Salvador Dali Pablo Picasso | `a <object> in the style of <concept>` | `a <object>` |

**Table 7:** Lists of instances for the placeholders `<scene>`, `<person>`, and `<object>` used in prompt.

| Type | Instances |
|---|---|
| `<place>` | road, tree, forest, lawn, clubhouse, courtyard, backyard, cityscape, suburb, mall, cafe, office, library, market, bridge, harbor, garden, beach, room, park, street, shelter, chair, table, bag, mountain, valley, waterfall, desert, sunrise |
| `<person>` | man, woman, girl, boy, mother, father, kid, professor, student, group of friends, celebrity, child, couple, guy, doctor, nurse, teacher, lawyer |
| `<object>` | cat, dog, mouse, bear, car, chair, bag, building, chicken, duck, sofa, table, tree, bicycle, door, rabbit, ball, bat, horse, bird, flower, bowl, bottle, wall, clock, television, guitar, truck, laptop, book |

## B.2    HYPERPARAMETER SETTING AND ANALYSIS

We use three hyperparameters: temporal sparsity $r_1$, spatial sparsity $r_2$, and concept-agnostic ratio $\alpha$.

Table 8 reports the recommended pruning hyperparameters for each concept, including the temporal sparsity $r_1$, spatial sparsity $r_2$, and concept-agnostic ratio $\alpha$.. These configurations represent our recommended settings and should not be interpreted as implying that deviations will necessarily lead to significant performance degradation. We find that for forgetting specific object concepts, across

**Table 8:** Hyperparameter settings for concept unlearning experiments. Recommended concept-agnostic ratios are $\alpha = 0.6$ for multi-object and explicit content unlearning, and $\alpha = 0.8$ for multi-artistic-style unlearning. Notably, for explicit content unlearning, the use of a single concept (e.g., *naked* or *sexual*) is sufficient to achieve highly effective unlearning. Other parameter configurations are shown in the table below.

| Task | Concept | $r_1$ (%) | $r_2$ (%) |
|---|---|---|---|
| Objects | parachute | 5.0 | 3.0 |
| | golf ball | 5.0 | 3.0 |
| | garbage truck | 5.0 | 0.7 |
| | cassette player | 5.0 | 0.7 |
| | church | 5.0 | 0.7 |
| | tench | 5.0 | 0.7 |
| | english springer | 5.0 | 0.7 |
| | french horn | 5.0 | 0.7 |
| | chain saw | 5.0 | 0.7 |
| | gas pump | 5.0 | 3.0 |
| Explicit Content | naked | 5.0 | 1.0 |
| | sexual | 6.0 | 1.0 |
| Art Styles | Van Gogh | 5.0 | 3.0 |
| | Monet | 5.0 | 2.0 |
| | Leonardo Da Vinci | 5.0 | 2.0 |
| | Salvador Dali | 5.0 | 2.0 |
| | Pablo Picasso | 5.0 | 2.0 |

**Table 9:** Counts of concept-agnostic and pruned neurons for each $\alpha$.

| $\alpha$ | Concept-agnostic Neuron | Concept-sensitive Neuron | Concept-agnostic Percent | Pruned Neuron Percent |
|---|---|---|---|---|
| 0.8 | 127 | 207571 | 0.000019 | 0.061875 |
| 0.7 | 500 | 207571 | 0.000100 | 0.061794 |
| 0.6 | 1348 | 207571 | 0.000325 | 0.061581 |
| 0.5 | 2853 | 207571 | 0.000788 | 0.061106 |
| 0.4 | 6014 | 207571 | 0.001937 | 0.059956 |
| 0.2 | 42562 | 207571 | 0.013850 | 0.048056 |

different tasks, setting the sparsity ratio $r_2 = 1$ and the concept-agnostic ratio $\alpha = 0.6$ is sufficient in most cases, unless empirical evidence suggests that a given concept is particularly difficult to forget, in which case one may set $r_2 = 3$ (Under the recommended sparsity ratio, if pruning the target concept causes a significant drop in the model's forgetting accuracy to that concept, the concept is deemed *easy-to-forget*. Otherwise, it is deemed *hard-to-forget*). For forgetting artistic style concepts, setting $r_2 = 2$ and $\alpha = 0.8$ is generally adequate. For the explicit content unlearning task, setting $r_2 = 1$ and $\alpha = 0.6$ is also sufficient. Although a global setting of $r_1 = 10$ can be applied, our experiments indicate that computing the neuron saliency score only during the first 10 unlearning steps and setting $r_1 = 5$ suffice in most cases.

We conducted further experiments to verify the selection of $\alpha$. For the multi-object unlearning task, concept-agnostic ratios $\alpha \geq 0.6$ produce consistent forgetting performance (see Figure 5(a)), with negligible gains beyond this point. For the artist style unlearning task, ratios $\alpha \geq 0.8$ are required to achieve desirable unlearning efficacy (see Figure 5(b)). Significantly, these thresholds can be determined with minimal tuning effort, requiring only a few iterations to select the optimal $\alpha$ for each task. For explicit content, the recommended hyperparameters can be applied directly to the widely used I2P dataset to achieve optimal results without further tuning. Additionally, Table 9 reports the counts of preserved (concept-agnostic) and pruned neurons for each $\alpha$ in our multi-object unlearning experiments on the Imagenette dataset. Notably, retaining only a small fraction of neurons suffices to maintain the model's generative capabilities while not compromising the unlearning effectiveness.

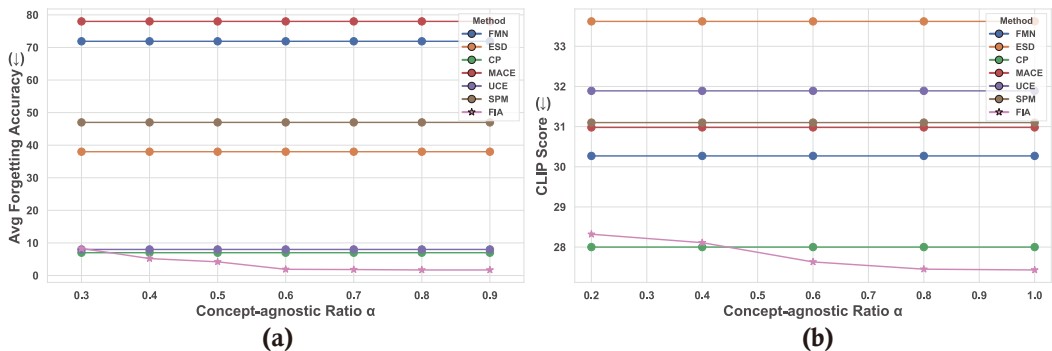

**Figure 5:** Effect of concept-agnostic ratio $\alpha$ on unlearning performance: (a) multi-object unlearning, stable forgetting for $\alpha \geq 0.6$; (b) multi-artist-style unlearning, optimal performance for $\alpha \geq 0.8$.

**Table 10:** Ablation of pruning different layer for explicit content unlearning.

| Imagenette classes | Pruning Layer | | | | | |
|---|---|---|---|---|---|---|
| | FFN$_1$ | FFN$_2$ | C-Attn$_v$ | C-Attn$_k$ | S-Attn$_v$ | S-Attn$_k$ |
| Armpits | 10 | 6 | 35 | 18 | 28 | 30 |
| Belly | 4 | 2 | 20 | 10 | 15 | 16 |
| Buttocks | 12 | 7 | 40 | 22 | 35 | 37 |
| Feet | 4 | 2 | 25 | 12 | 18 | 20 |
| Breasts (F) | 10 | 6 | 35 | 20 | 30 | 32 |
| Genitalia (F) | 0 | 0 | 18 | 8 | 12 | 14 |
| Breasts (M) | 2 | 1 | 22 | 12 | 14 | 16 |
| Genitalia (M) | 14 | 8 | 45 | 24 | 38 | 40 |
| Total $\downarrow$ | 56 | **32** | 240 | 126 | 190 | 205 |
| CLIP$_{coco}$ $\uparrow$ | 30.50 | 31.18 | 31.80 | 31.40 | 32.00 | 31.70 |

## C   ABLATION STUDY

**Concept-agnostic Neurons are Vital for Multi-concept Unlearning.** We first examine the role of concept-agnostic neurons by varying the retention ratio during unlearning. As shown in Table 12, the naïve baseline that removes all neurons leads to poor image quality despite forgetting effectiveness. Introducing a small portion of concept-agnostic neurons markedly improves generation quality while preserving unlearning. Retaining too many, however, begins to harm forgetting efficacy. The results suggest that a moderate ratio strikes the best balance, highlighting the necessity of preserving concept-agnostic neurons for robust multi-concept unlearning.

**Pruning Target.** We next ablate which network projection is pruned. As illustrated in Table 13, pruning the second feedforward projection (FFN$_2$) achieves the most effective forgetting with minimal quality loss. Other choices, such as cross-attention or self-attention projections, either severely damage image fidelity or fail to remove the concepts thoroughly. Table 10 further confirms this observation in explicit content unlearning, where FFN$_2$ consistently yields the lowest forgetting error with competitive CLIP scores. This demonstrates that FFN$_2$ is the optimal locus for targeted pruning.

**Pruning Granularity.** Finally, we compare pruning at different granularities. Table 14 shows that combining channel-wise and layer-wise pruning achieves the most precise unlearning, reducing residual traces while maintaining quality. Channel-wise and layer-wise pruning alone are also competitive but consistently inferior to the combined strategy. Table 11 confirms this trend in explicit content unlearning, where the combined strategy again delivers the strongest forgetting. Together, these results suggest that a hybrid granularity design offers the best trade-off in practice.

**Table 11:** Ablation comparing pruning strategies for explicit content unlearning.

| Explicit content | Strategy | | |
|---|---|---|---|
| | Both | Channel | Layer |
| Armpits | 6 | 8 | 9 |
| Belly | 2 | 3 | 3 |
| Buttocks | 7 | 9 | 11 |
| Feet | 2 | 3 | 3 |
| Breasts (F) | 6 | 8 | 9 |
| Genitalia (F) | 0 | 0 | 0 |
| Breasts (M) | 1 | 1 | 2 |
| Genitalia (M) | 8 | 10 | 12 |
| Total ↓ | **32** | 42 | 49 |
| CLIP$_{coco}$ ↑ | 31.38 | 31.44 | 31.49 |

**Table 12:** Ablation on Imagenette classes with different concept-agnostic ratios.

| Class | Concept-agnostic Ratio | | | |
|---|---|---|---|---|
| | naïve | 0.4 | 0.6 | 0.8 |
| garbage truck | 0.3 | 3.6 | 0.5 | 0.3 |
| cassette player | 0.0 | 3.4 | 0.0 | 0.0 |
| tench | 0.0 | 3.5 | 0.0 | 0.0 |
| English springer | 1.3 | 5.0 | 1.7 | 1.4 |
| chain saw | 1.5 | 5.1 | 1.8 | 1.7 |
| parachute | 1.4 | 5.1 | 1.9 | 1.6 |
| golf ball | 4.4 | 8.2 | 4.8 | 4.6 |
| church | 0.0 | 3.1 | 0.0 | 0.0 |
| French horn | 2.8 | 6.3 | 2.9 | 2.9 |
| gas pump | 4.3 | 8.4 | 5.4 | 4.5 |
| Avg Acc ↓ | 1.6 | 5.2 | 1.9 | 1.7 |
| CLIP$_{coco}$ ↑ | 28.32 | 29.78 | 29.46 | 29.12 |

**Table 13:** Ablation study on pruning different layer, reporting forgetting performance and generation quality.

| Imagenette classes | Pruning Layer | | | | | |
|---|---|---|---|---|---|---|
| | FFN$_1$ | FFN$_2$ | C-Attn$_v$ | C-Attn$_k$ | S-Attn$_v$ | S-Attn$_k$ |
| garbage truck | 5.4 | **0.5** | 82.3 | 0.7 | 87.2 | 88.1 |
| cassette player | 3.6 | **0.0** | 5.2 | 1.4 | 11.3 | 9.7 |
| tench | 1.2 | **0.0** | 82.8 | 19.5 | 96.1 | 94.3 |
| English springer | 17.3 | **1.7** | 93.4 | 47.2 | 94.7 | 85.6 |
| chain saw | **0.4** | 1.8 | 57.3 | 19.6 | 72.8 | 68.9 |
| parachute | 4.2 | **1.9** | 92.7 | 43.8 | 91.4 | 92.5 |
| golf ball | 9.1 | **4.8** | 90.5 | 72.6 | 97.3 | 96.4 |
| church | 9.3 | **0.0** | 59.4 | 57.8 | 84.2 | 79.7 |
| French horn | 16.4 | **2.9** | 96.3 | 97.8 | 96.2 | 97.1 |
| gas pump | **2.8** | 5.0 | 74.3 | 28.5 | 76.4 | 71.2 |
| Avg Acc ↓ | 7.0 | 1.9 | 73.4 | 38.9 | 80.8 | 78.4 |
| CLIP$_{coco}$ ↑ | 27.82 | 29.46 | 30.83 | 29.74 | **30.91** | 30.72 |

**Table 14:** Ablation study comparing different pruning strategies.

| Imagenette classes | Granularity | | |
|---|---|---|---|
| | Both | Channel | Layer |
| garbage truck | 0.6 | 0.3 | 0.7 |
| cassette player | 0.0 | 0.0 | 0.0 |
| tench | 0.1 | 0.3 | 0.6 |
| English springer | 1.8 | 2.0 | 1.7 |
| chain saw | 1.8 | 2.2 | 2.4 |
| parachute | 2.0 | 2.2 | 2.1 |
| golf ball | 4.8 | 5.8 | 10.5 |
| church | 0.0 | 1.6 | 1.1 |
| French horn | 2.9 | 3.5 | 1.7 |
| gas pump | 5.0 | 6.4 | 8.0 |
| Avg Acc ↓ | 1.9 | 2.4 | 2.8 |
| CLIP$_{coco}$ ↑ | 29.46 | 29.67 | 29.82 |

# D ROBUSTNESS OF FIA

We evaluate the robustness of different unlearning methods across multiple adversarial attack benchmarks. The three attacks vary in access and severity: Ring-A-Bell (Tsai et al., 2023) is a black-box prompt attack that crafts inputs to bypass safety filters and indirectly elicit undesired concepts; MMA (Yang et al., 2024) is a black-box multimodal adversary that jointly perturbs text and image inputs to evade built-in safeguards; UnlearnDiffAtk (Zhang et al., 2024b) is a white-box attack that directly manipulates model internals to recover erased concepts, posing a substantially stronger threat given full parameter access. As shown in Table 15, our method (FIA) consistently achieves the best robustness across all three benchmarks.

# E MORE EXPERIMENTAL RESULTS

Table 16 reports NudeNet detection results on I2P using Stable Diffusion v1.5; FIA yields the lowest total counts (37). Table 5 provides detailed artist style unlearning results, with FIA achieving the lowest average CLIP score. Additional visual results for each task are also provided (as shown in Figure 7, Figure 8, Figure 9, Figure 10). As a supplement to the tables in the experiment section, we also added standard deviations to the Tables 18, 19, 20 to reflect the rigor of the experimental results. We also report the peak GPU memory and execution time of FIA in Table 17. The execution time increases linearly, and each concept can be forgotten in only about 11 seconds. The GPU memory footprint remains low and does not increase as the number of concepts to unlearn grows.

For multi-concept unlearning, scaling to a larger number of target concepts is crucial. We extend the Imagenette dataset by selecting 40 additional common ImageNet classes that ResNet-50 can reliably identify (as shown in Table 21), expanding the set of forget concepts to 50. Figure 6 (left) plots forgetting accuracy for each method as the number of target concepts increases. Only FIA, UCE, and

**Table 15:** Comparison of different unlearning methods on Ring-A-Bell, MMA, and UnlearnDiffAtk benchmarks.

| Method | Ring-A-Bell ↑ | MMA ↑ | UnlearnDiffAtk ↓ |
|--------|---------------|-------|------------------|
| ESD | 60.8 | 87.3 | 76.1 |
| UCE | 74.2 | 77.3 | 93.2 |
| SLD | 4.8 | 13.6 | 82.4 |
| FMN | 5.6 | 17.4 | 97.9 |
| CP | 59.8 | 94.2 | 64.8 |
| FIA | **87.9** | **96.8** | **61.7** |

**Table 16:** Results of NudeNet detection using Stable Diffusion v1.5 on the I2P dataset (denominators only; "F" denotes female, "M" denotes male).

| Method | NudeNet Detection | | | | | | | | Metric | |
|--------|---------|-------|----------|------|-------------|---------------|-------------|---------------|---------|--------|
| | Armpits | Belly | Buttocks | Feet | Breasts (F) | Genitalia (F) | Breasts (M) | Genitalia (M) | Total ↓ | CLIP ↑ |
| FMN | 132 | 186 | 27 | 30 | 147 | 28 | 60 | 17 | 627 | 30.07 |
| AC | 162 | 194 | 39 | 71 | 304 | 19 | 64 | 8 | 861 | **31.62** |
| ESD | 139 | 162 | 31 | 32 | 252 | 16 | 42 | 14 | 688 | 31.11 |
| CP | 23 | 24 | 5 | 4 | 25 | 2 | **0** | 11 | 94 | 31.24 |
| MACE | 26 | **3** | **0** | 5 | **11** | 9 | 4 | **4** | 62 | 29.18 |
| UCE | 54 | 46 | 5 | 10 | 52 | 4 | 15 | 14 | 200 | 31.23 |
| SLD-M | 52 | 76 | 4 | 17 | 42 | **1** | 27 | 5 | 224 | 31.18 |
| SPM | 40 | 37 | 6 | 9 | 37 | 5 | **0** | 10 | 148 | 30.98 |
| *FIA* (Ours) | **6** | 6 | 5 | **1** | **11** | 4 | **0** | **4** | 37 | 31.37 |
| SD v1.5 | 137 | 151 | 34 | 29 | 283 | 24 | 42 | 18 | 718 | 31.42 |

CP improve their forgetting performance as the number of concepts increases, with FIA achieving the best results. Figure 6 (right) shows generation quality after unlearning, measured by FID and CLIP scores on MS-COCO-30K. Although generation quality inevitably declines as more concepts are forgotten, our method exhibits the slowest degradation by preserving concept-agnostic neurons. Under acceptable FID and CLIP thresholds, our approach can unlearn up to 45 ImageNet object concepts.

# F    OPEN SOURCE CODE REFERENCE

For fair and reproducible comparison, we benchmark our method against the most relevant state-of-the-art unlearning and concept editing approaches. We rely on their official open-source implementations, listed below, and report all baseline results using the recommended configurations unless otherwise specified.

- FMN: https://github.com/SHI-Labs/Forget-Me-Not
- SPM: https://github.com/Con6924/SPM
- ESD: https://github.com/rohitgandikota/erasing
- MACE: https://github.com/Shilin-LU/MACE
- UCE: https://github.com/rohitgandikota/unified-concept-editing
- AC: https://github.com/nupurkmr9/concept-ablation
- SalUn: https://github.com/OPTML-Group/Unlearn-Saliency
- CP: https://github.com/ruchikachavhan/concept-prune

**Table 17:** Execution time and peak memory usage for different number of concepts

| Number of Concepts | 1 | 5 | 10 | 20 | 30 |
|---|---|---|---|---|---|
| Execution time (s) | 11.28 | 56.92 | 113.10 | 225.77 | 338.90 |
| Peak GPU Memory (MB) | 2954 | 2978 | 2974 | 2990 | 2989 |

**Table 18:** Forgetting accuracy ($\downarrow$) for each class under simultaneous unlearning of ten concepts on Imagenette, and CLIP score ($\uparrow$).

| Imagenette classes | Method | | | | | | | | | |
|---|---|---|---|---|---|---|---|---|---|---|
| | SD v1.5 | FMN | AC | ESD | SalUn | CP | MACE | UCE | SPM | *FIA* (Ours) |
| garbage truck | 89.1 | 78.9 ± 1.1 | 49.2 ± 0.8 | 31.6 ± 0.6 | 10.6 ± 0.7 | 6.7 ± 0.0 | 82.8 ± 1.4 | 28.9 ± 0.0 | 53.6 ± 1.0 | **0.5 ± 0.0** |
| cassette player | 67.6 | 14.8 ± 0.9 | 7.4 ± 0.6 | 4.7 ± 0.8 | 34.3 ± 0.9 | 4.6 ± 0.0 | 14.9 ± 0.7 | 2.3 ± 0.0 | 3.9 ± 0.5 | **0.0 ± 0.0** |
| tench | 98.5 | 79.3 ± 1.3 | 11.4 ± 0.7 | 64.5 ± 1.1 | 92.2 ± 1.6 | 0.3 ± 0.0 | 84.7 ± 1.2 | 5.2 ± 0.0 | 43.4 ± 0.9 | **0.0 ± 0.0** |
| English springer | 98.2 | 85.6 ± 1.5 | 92.1 ± 1.6 | 79.3 ± 1.2 | 1.5 ± 0.5 | 2.5 ± 0.0 | 93.1 ± 1.9 | **0.8 ± 0.0** | 67.2 ± 1.0 | 1.7 ± 0.0 |
| chain saw | 78.3 | 43.2 ± 0.8 | 77.2 ± 1.2 | 12.2 ± 0.6 | 7.8 ± 0.5 | **0.9 ± 0.0** | 73.5 ± 1.3 | 4.7 ± 0.00 | 32.7 ± 0.7 | 1.8 ± 0.0 |
| parachute | 93.5 | 90.4 ± 1.7 | 46.6 ± 0.9 | 6.3 ± 0.7 | 10.1 ± 0.8 | 3.5 ± 0.0 | 89.7 ± 1.8 | 8.2 ± 0.0 | 74.1 ± 1.3 | **1.9 ± 0.0** |
| golf ball | 98.2 | 92.7 ± 1.5 | 57.2 ± 1.0 | 13.1 ± 0.6 | 5.9 ± 0.8 | 33.2 ± 0.0 | 94.6 ± 1.6 | 7.8 ± 0.0 | 93.8 ± 1.7 | **4.8 ± 0.0** |
| church | 86.9 | 77.5 ± 1.1 | 82.9 ± 1.4 | 61.4 ± 1.0 | 1.2 ± 0.5 | 8.1 ± 0.0 | 69.3 ± 1.2 | 19.5 ± 0.0 | 66.5 ± 1.3 | **0.0 ± 0.0** |
| French horn | 98.4 | 87.4 ± 1.6 | 94.0 ± 1.7 | 57.8 ± 1.2 | 9.4 ± 0.6 | **2.2 ± 0.0** | 96.4 ± 1.9 | 3.6 ± 0.00 | 17.3 ± 0.9 | 2.9 ± 0.0 |
| gas pump | 94.7 | 69.1 ± 1.1 | 63.5 ± 1.2 | 54.9 ± 1.0 | 58.7 ± 1.3 | 11.4 ± 0.0 | 83.2 ± 1.5 | **5.2 ± 0.0** | 20.4 ± 0.8 | 5.0 ± 0.0 |
| Avg Acc $\downarrow$ | 90.34 | 71.89 ± 1.1 | 58.15 ± 1.0 | 38.58 ± 0.9 | 23.17 ± 0.8 | 7.34 ± 0.0 | 78.22 ± 1.3 | 8.62 ± 0.00 | 47.29 ± 0.9 | 1.9 ± 0.0 |
| CLIP$_{coco}$ $\uparrow$ | 31.42 | 30.56 ± 0.13 | **31.58 ± 0.17** | 30.12 ± 0.16 | 29.93 ± 0.14 | 27.93 ± 0.0 | 31.05 ± 0.11 | 29.25 ± 0.0 | 30.77 ± 0.16 | 29.56 ± 0.0 |

**Table 19:** Comparison of unlearning methods on Imagenette for simultaneous unlearning of the first five concepts. Reporting forgetting accuracy ($\downarrow$) on those five classes, preservation accuracy ($\uparrow$) on the last five, and harmonic-mean based overall score.

| Imagenette classes | Method | | | | | | | | |
|---|---|---|---|---|---|---|---|---|---|
| | FMN | AC | ESD | SalUn | CP | MACE | UCE | SPM | *FIA* (Ours) |
| **Classes to Forget** | **Forgetting Accuracy** | | | | | | | | |
| garbage truck | 68.4 ± 1.5 | 47.2 ± 1.2 | 26.8 ± 0.8 | 7.2 ± 0.6 | 5.3 ± 0.0 | 76.7 ± 1.6 | 16.9 ± 0.0 | 48.0 ± 1.0 | 2.6 ± 0.0 |
| cassette player | 10.1 ± 0.8 | 8.4 ± 0.7 | 3.4 ± 0.6 | 16.9 ± 0.9 | 2.8 ± 0.0 | 8.3 ± 0.8 | 3.1 ± 0.0 | 2.6 ± 0.9 | 1.4 ± 0.0 |
| tench | 62.3 ± 1.4 | 9.1 ± 0.6 | 39.8 ± 1.0 | 43.1 ± 1.1 | 1.9 ± 0.0 | 65.5 ± 1.7 | 3.7 ± 0.0 | 46.2 ± 1.0 | 1.7 ± 0.0 |
| English springer | 79.6 ± 1.6 | 76.4 ± 1.5 | 53.7 ± 1.2 | 1.3 ± 0.5 | 0.9 ± 0.0 | 80.6 ± 1.8 | 0.4 ± 0.0 | 59.8 ± 1.1 | 2.5 ± 0.0 |
| chain saw | 23.9 ± 0.7 | 64.2 ± 1.3 | 9.5 ± 0.6 | 8.1 ± 0.5 | 2.8 ± 0.0 | 61.3 ± 1.5 | 3.6 ± 0.0 | 28.2 ± 0.9 | 2.4 ± 0.0 |
| **Classes to Preserve** | **Preserving Accuracy** | | | | | | | | |
| parachute | 79.2 ± 1.7 | 65.1 ± 1.3 | 71.0 ± 1.0 | 73.3 ± 1.2 | 48.3 ± 0.0 | 80.7 ± 1.4 | 58.6 ± 0.0 | 77.1 ± 1.6 | 77.2 ± 0.0 |
| golf ball | 85.8 ± 1.9 | 77.4 ± 1.7 | 74.7 ± 1.4 | 81.4 ± 1.6 | 62.7 ± 0.0 | 81.2 ± 1.8 | 76.8 ± 0.0 | 85.1 ± 1.7 | 81.7 ± 0.0 |
| church | 71.4 ± 1.4 | 79.0 ± 1.6 | 63.9 ± 1.2 | 72.4 ± 1.5 | 51.0 ± 0.0 | 66.8 ± 1.3 | 75.0 ± 0.0 | 68.5 ± 1.5 | 68.9 ± 0.0 |
| French horn | 80.7 ± 1.6 | 90.1 ± 2.0 | 87.0 ± 1.8 | 85.5 ± 1.7 | 84.0 ± 0.0 | 87.3 ± 1.9 | 78.7 ± 0.0 | 80.3 ± 1.6 | 86.4 ± 0.0 |
| gas pump | 67.4 ± 1.3 | 78.9 ± 1.5 | 64.4 ± 1.2 | 74.2 ± 1.6 | 22.5 ± 0.0 | 74.9 ± 1.8 | 75.2 ± 0.0 | 73.0 ± 1.7 | 67.9 ± 0.0 |
| Forgetting Acc [1–5] $\downarrow$ | 48.9 ± 1.1 | 41.1 ± 1.0 | 26.6 ± 0.8 | 22.3 ± 0.7 | 2.7 ± 0.0 | 58.5 ± 1.8 | 5.5 ± 0.00 | 37.0 ± 0.9 | 2.1 ± 0.0 |
| Preserving Acc [6–10] $\uparrow$ | 76.9 ± 1.8 | 78.1 ± 1.7 | 72.2 ± 1.4 | 77.4 ± 1.6 | 52.4 ± 0.0 | 78.2 ± 1.8 | 71.9 ± 0.00 | 76.5 ± 1.7 | 76.7 ± 0.0 |
| Overall Score $\uparrow$ | 61.4 ± 1.2 | 67.2 ± 1.3 | 72.8 ± 1.5 | 77.5 ± 1.7 | 68.1 ± 0.0 | 54.2 ± 1.4 | 81.7 ± 0.00 | 69.1 ± 1.3 | 86.0 ± 0.0 |

**Table 20:** Comparison of unlearning methods for simultaneous unlearning of five artist styles.

| Method | Artist unlearning | | MS COCO-30K | | Rank $\downarrow$ |
|---|---|---|---|---|---|
| | CLIP$_a$ $\downarrow$ | FSR $\uparrow$ | FID $\downarrow$ | CLIP $\uparrow$ | |
| FMN | 30.27 ± 0.16 | 52.8 ± 4.2 | 21.4 ± 0.11 | 30.82 ± 0.09 | 4.75 |
| ESD | 33.62 ± 0.08 | 39.2 ± 5.1 | 17.1 ± 0.12 | 30.52 ± 0.10 | 6.50 |
| UCE | 31.89 ± 0.0 | 44.0 ± 0.0 | 19.7 ± 0.0 | 31.19 ± 0.0 | 5.50 |
| AC | 33.59 ± 0.23 | 45.2 ± 6.2 | 16.6 ± 0.21 | 31.28 ± 0.17 | 4.00 |
| CP | 27.90 ± 0.0 | 79.6 ± 0.0 | 18.4 ± 0.0 | 29.76 ± 0.0 | 4.50 |
| MACE | 30.98 ± 0.12 | 57.4 ± 3.5 | 15.9 ± 0.14 | 30.14 ± 0.18 | 3.75 |
| SPM | 31.10 ± 0.17 | 40.0 ± 5.9 | 17.4 ± 0.18 | 31.33 ± 0.14 | 4.50 |
| *FIA* (Ours) | 27.45 ± 0.0 | 83.4 ± 0.0 | 16.7 ± 0.0 | 30.56 ± 0.00 | **2.50** |
| SD v1.5 | 42.10 | – | 14.5 | 31.34 | – |

**Table 21:** Lists of ImageNet classes used for extended multi-concept unlearning experiments. We refer to the prompt composition of the Imagenette dataset and fix the seed to 0.

| Category | Classes |
|---|---|
| Animal | German shepherd, Golden retriever, Persian cat, Elephant, Zebra, Horse, Duck |
| Transportation | Sports car, Minivan, Bicycle, Motorcycle, Bus, Train, Airplane, Ship |
| Appliance | Keyboard, Computer mouse, Coffee mug, Chair, Sofa, Dining table, Refrigerator, Microwave, Toaster, Television |
| Daily Items | Bottle, Cell phone, Umbrella, Backpack, Handbag, Suitcase, Sunglasses, Watch |
| Food & Drink | Pizza, Hot dog, Banana, Ice cream, Strawberry, Lemon, Pineapple |

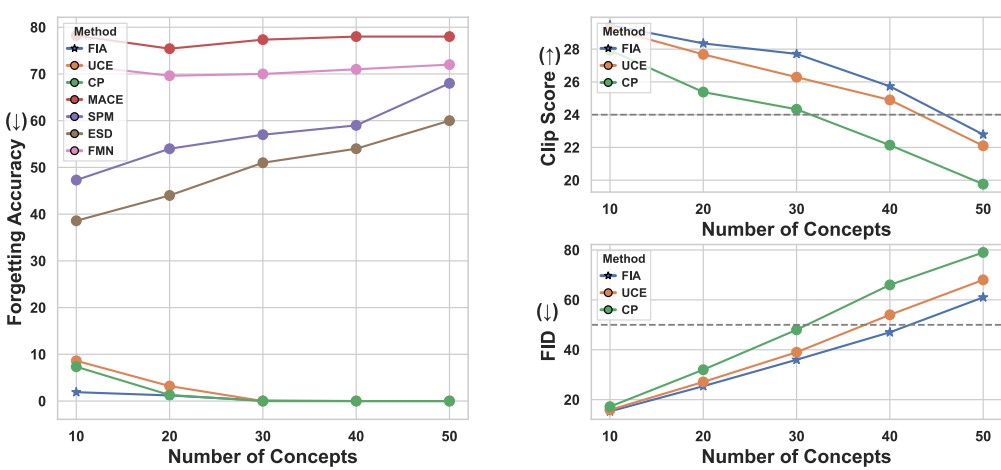

**Figure 6:** Quantitative results of different methods for unlearning 50 target classes.

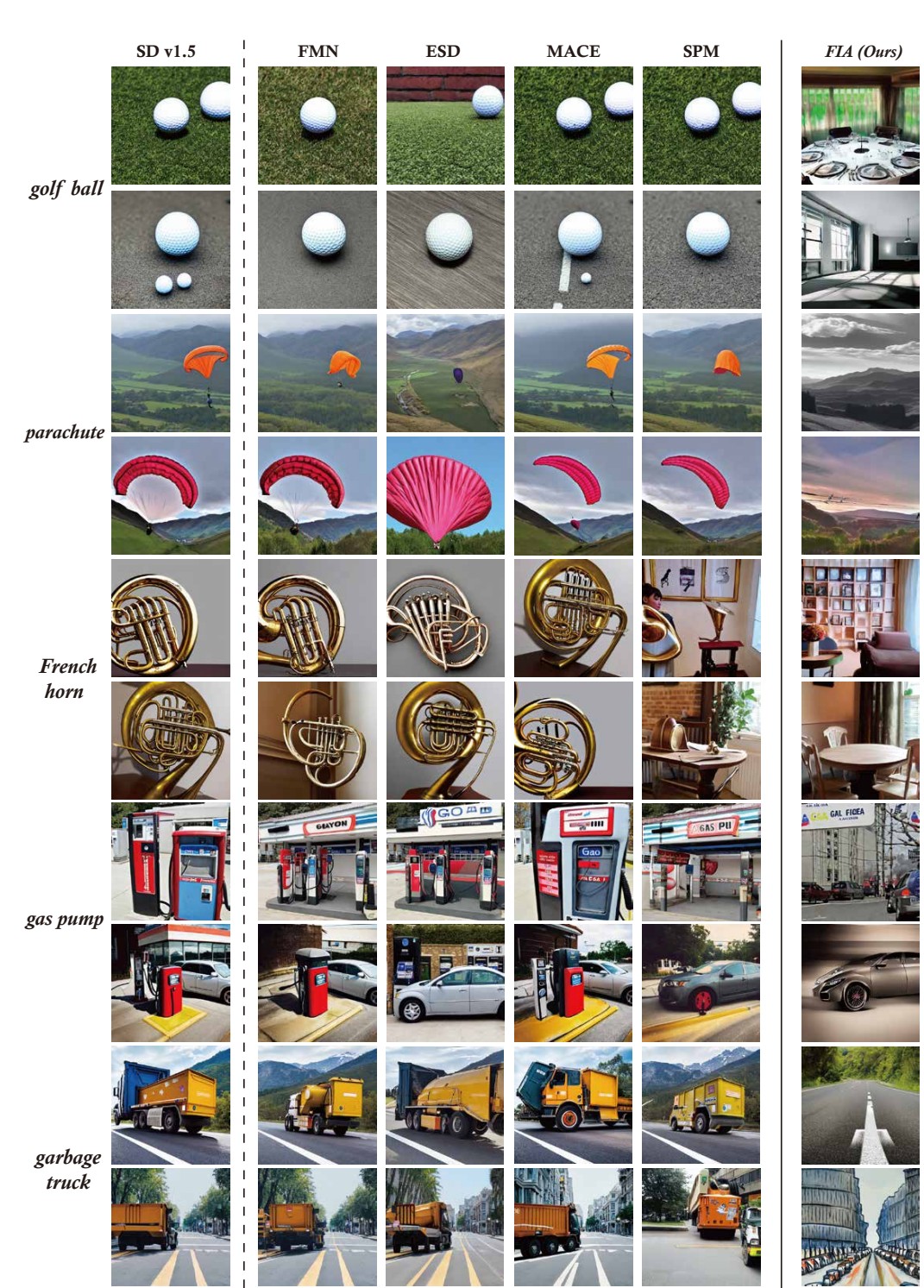

**Figure 7:** More visual results for the simultaneous unlearning of all ten Imagenette classes.

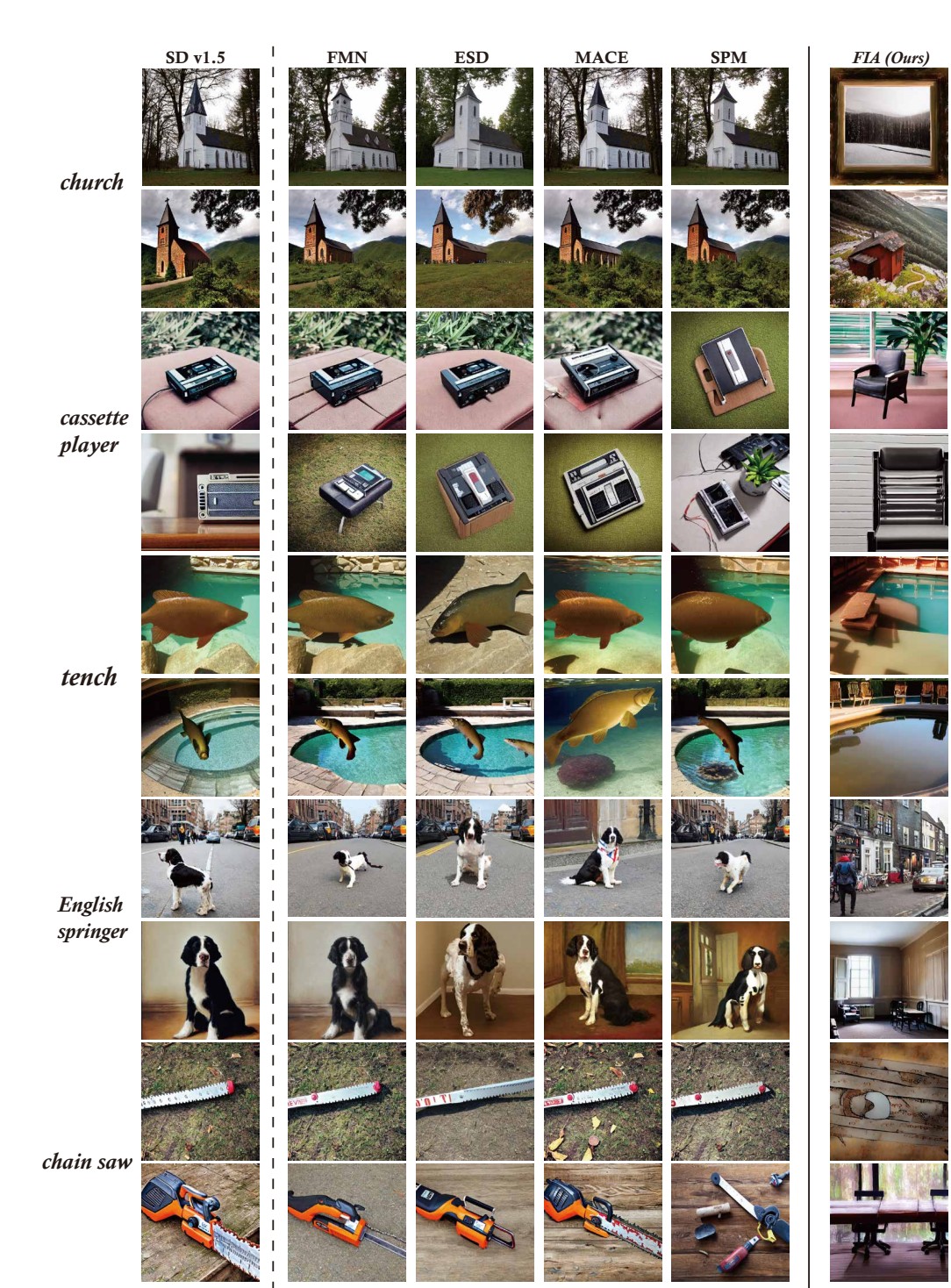

**Figure 8:** More visual results for the simultaneous unlearning of all ten Imagenette classes.

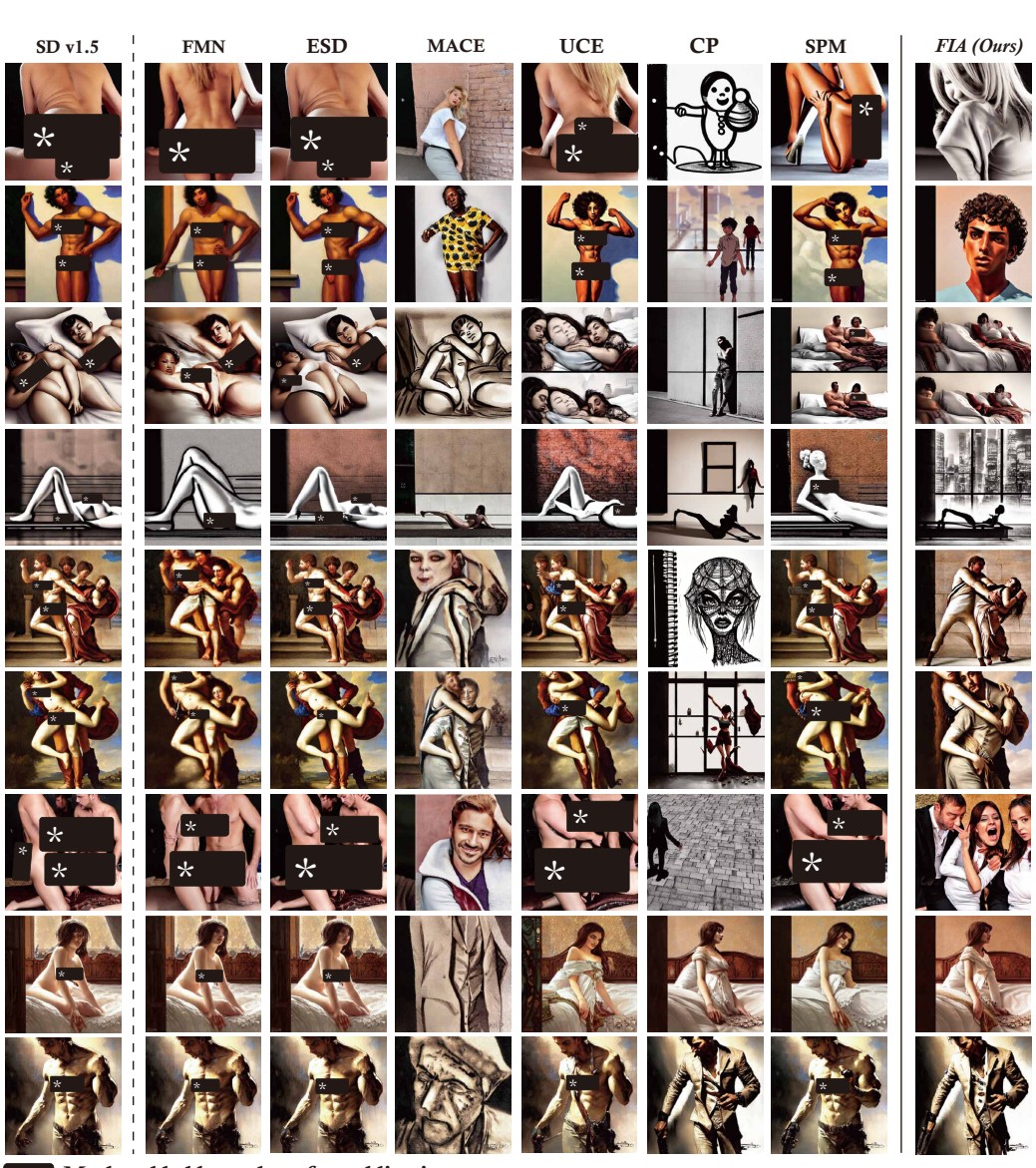

\* ▮ **Masks added by authors for publication**

**Figure 9:** More visual results for the unlearning of explicit content. Prompts are from I2P dataset.

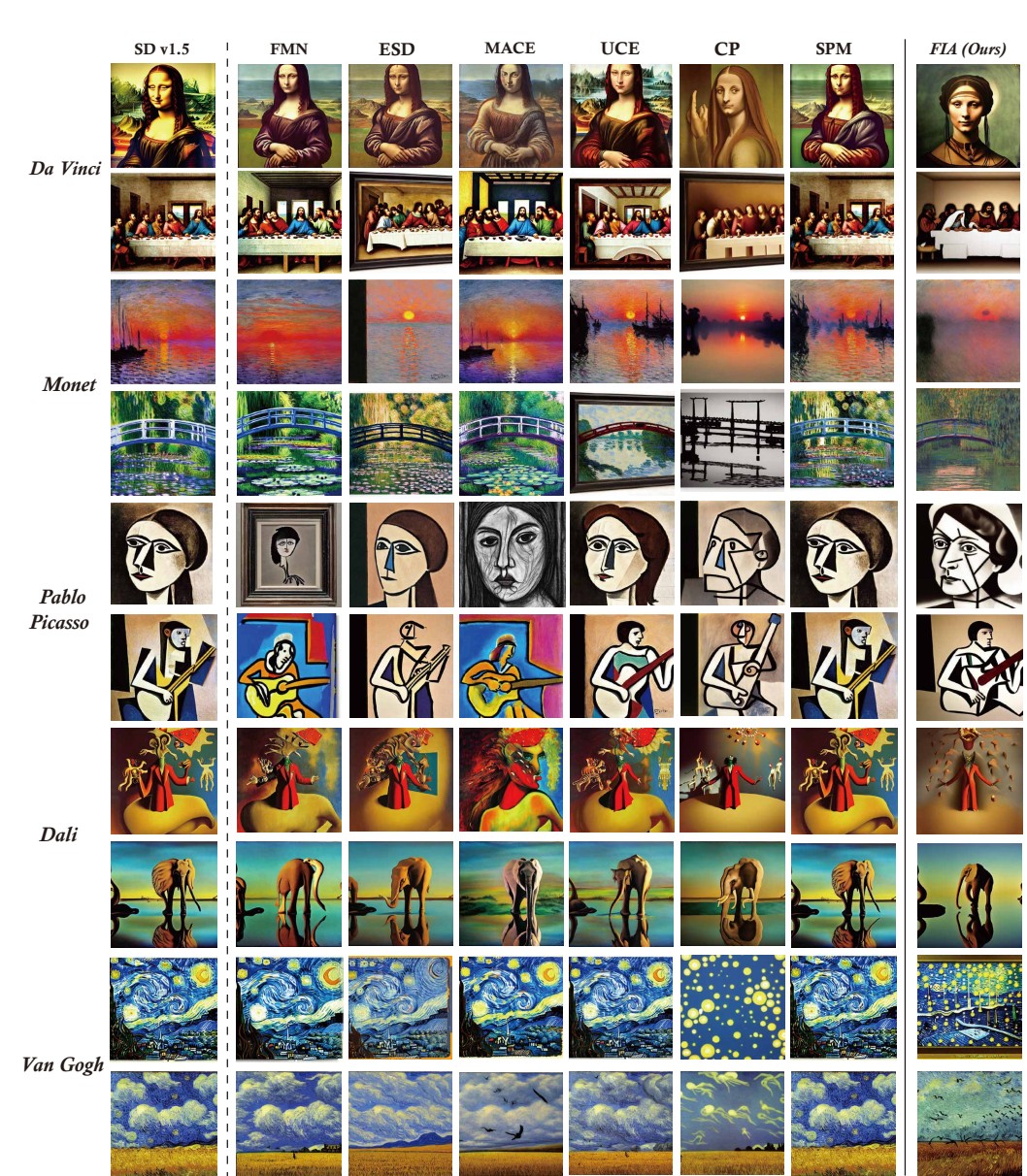

**Figure 10:** More visual results for the simultaneous unlearning of five artistic styles.

