# OpenReview forum: "Forget-It-All: Multi-Concept Machine Unlearning via Concept-Aware Neuron Masking"
_ICLR.cc/2026/Conference — Submitted to ICLR 2026_

### Official Review · Reviewer_fm5g · 2025-10-26

**Soundness:** 2
**Presentation:** 3
**Contribution:** 3
**Rating:** 6
**Confidence:** 3

**Summary:**

This paper introduces Forget-It-All, a training-free framework for multi-concept machine unlearning in text-to-image diffusion models. FIA computes Contrastive Concept Saliency to quantify each weight's contribution to target concepts, identifies Concept-Sensitive Neurons through temporal and spatial aggregation, and constructs masks that preserve Concept-Agnostic Neurons (which respond broadly across concepts) while pruning concept-specific neurons. The method is evaluated on three unlearning tasks (multi-object, explicit content, artistic style) using Stable Diffusion, achieving SOTA forgetting performance, good generation quality, at under 0.3% overall sparsity.

**Strengths:**

- To my knowledge, the paper presents the first work to explicitly connect model sparsity with multi-concept unlearning through unstructured neuron masking. The introduction of concept-agnostic neurons is a particularly insightful contribution.
- FIA operates training-free, unlike most competing methods. This makes the method efficient and less prone to overfitting or hyperparameter sensitivity.
- The paper includes experiments across distinct unlearning tasks demonstrating consistent state-of-the-art performance. The forgetting accuracy results are particularly impressive (e.g., Table 1, 1.9% on Imagenette).
- The three-component approach (Contrastive Concept Saliency, Concept-Sensitive Neurons, and mask fusion with concept-agnostic neurons) is systematic and well-motivated.
- The paper provides detailed ablations on key design choices including concept-agnostic ratio, pruning target, and pruning granularity (Section C), validating the importance of each component.
- The paper shows the method can scale to unlearning up to 50 concepts (Appendix E), which is important for practical applications.

**Weaknesses:**

- The paper lacks theoretical connections to related continual learning frameworks, particularly Elastic Weight Consolidation (EWC) [1] and related Fisher Information-based approaches.
- The Contrastive Concept Saliency formulation (Eq. 1) combines weight magnitude, activation norm, and cosine similarity, but the paper doesn't provide theoretical justification for this specific combination or discuss its relationship to established importance metrics in the literature.
- While the paper reports CLIP scores on MS COCO-30K for most experiments, FID scores are not consistently reported across all tasks. Table 1 (multi-object unlearning of 10 concepts) and Table 2 only show CLIP scores, while Table 3 (explicit content) and Table 4 (artistic style) include both FID and CLIP. This inconsistency makes it difficult to fully assess generation quality degradation across different unlearning scenarios.
- The experiments are restricted to Stable Diffusion v1.4 and v1.5, which are relatively older architectures. The paper does not evaluate on more recent models such as Stable Diffusion XL [2], Stable Diffusion 3 [3], or orther transformer-based architectures like FLUX.1 [4] or Sana [5]. This limits the generalizability claims of the approach.
- Section 3.1 introduces concept prompts and base prompts but doesn't specify how many prompts are used for each concept, whether prompts are manually designed or automatically generated, and how sensitive the method is to prompt variations. While Appendix B.1 provides templates, the number of instances and the variance in results across different prompt sets are not reported.
- While Table 17 reports execution time and memory for FIA, similar metrics are not provided for baseline methods, making it difficult to assess the true computational advantage. The paper claims efficiency but doesn't quantify the training time and resources required by fine-tuning-based baselines for fair comparison.
- The paper doesn't adequately discuss when the method fails or performs suboptimally. For instance, in Table 1, some classes like "golf ball" have higher forgetting accuracy (4.8%) compared to others. Understanding why certain concepts are harder to forget would strengthen the paper.

In conclusion, the paper presents a novel and effective approach with strong empirical results and comprehensive experiments. However, due to the lack of theoretical grounding (particularly connections to EWC and continual learning literature) and limited architectural coverage, among the rest of the issues raised, I am inclined to lower my score.


[1] Kirkpatrick, James, et al. "Overcoming catastrophic forgetting in neural networks." Proceedings of the national academy of sciences 114.13 (2017): 3521-3526.
[2] Podell, Dustin, et al. "Sdxl: Improving latent diffusion models for high-resolution image synthesis." arXiv preprint arXiv:2307.01952 (2023).
[3] Esser, Patrick, et al. "Scaling rectified flow transformers for high-resolution image synthesis." ICML 2024.
[4] Black Forest Labs, "FLUX", 2024.
[5] XIE, Enze, et al. "Sana: Efficient high-resolution image synthesis with linear diffusion transformers." ICLR 2025.

**Questions:**

- What happens if you need to unlearn additional concepts after already applying FIA? Can the method be applied iteratively, or does this require recomputing masks from scratch?
- Under what conditions does FIA fail to adequately unlearn concepts? Are there specific types of concepts or concept combinations that are particularly challenging for your approach?

---

> ### Author Response · Authors · 2025-11-28
>
> # Rebuttal 1/3
>
> We sincerely thank the reviewer for the careful reading and constructive feedback, which significantly improved the clarity and reproducibility of our work.
>
> ### **Weakness 1. Comparison with Elastic Weight Consolidation.**
>
> We appreciate the reviewer’s suggestion regarding the theoretical comparison to continual learning frameworks such as Elastic Weight Consolidation (EWC), and we will include a more explicit discussion in the revised version. EWC and other Fisher information based methods address catastrophic forgetting in sequential task training by applying a weighted quadratic penalty to parameters that are important for previous tasks. In contrast, FIA does not learn a new task but instead focuses on selectively removing concept specific representations within a pretrained diffusion model. FIA introduces Contrastive Concept Saliency to identify neurons that show stable response differences between concept prompts and base prompts, and further refines these neurons using temporal consistency and spatial sparsity to obtain truly concept sensitive units. In multi concept settings, FIA also preserves concept agnostic neurons that are necessary for maintaining general generative quality, which cannot be captured by EWC’s quadratic penalty. Therefore, while EWC works at a task level using global curvature signals, FIA operates directly on fine grained concept level structures, enabling more precise removal and better quality retention in multi concept unlearning.
>
> ### **Weakness 2. Theoretical basis of FIA**
>
> Saliency-based pruning[1][2] provides a well-established foundation for identifying parameters that matter for representation flow and real data usage. Prior work such as Wanda[2] shows that simple activation-weighted magnitude naturally approximates second-order importance, grounding these measures as principled pruning criteria. Building on this view, our Contrastive Concept Saliency remains fully aligned with this theoretical lineage while introducing only the components necessary for precise concept attribution. We retain weight magnitude to capture structural capacity, activation strength to reflect actual usage, and a cosine-based term to measure whether a connection meaningfully participates in concept-related signal propagation. These factors together provide a compact, training-free estimate of concept-specific energy flow and allow us to isolate neurons that are genuinely tied to the target concept.
>
> [1] Sun M, Liu Z, Bair A, et al. A simple and effective pruning approach for large language models[J]. arXiv preprint arXiv:2306.11695, 2023.
>
> [2] Yin L, Wu Y, Zhang Z, et al. Outlier weighed layerwise sparsity (owl): A missing secret sauce for pruning llms t2 high sparsity[J]. arXiv preprint arXiv:2310.05175, 2023.
>
> ### **Weakness 3. FID report for Multi-concept experiment**
>
> To provide a more complete assessment of generation fidelity, we additionally computed FID for the 10-concept multi-object unlearning setting. As shown in the table below, although FIA achieves by far the strongest forgetting performance (Avg Acc = 1.9, substantially better than all baselines), its generation quality remains competitive across both CLIP and FID. This indicates that FIA does not introduce disproportionate degradation in either semantic alignment (CLIP) or distribution-level fidelity (FID). In other words, FIA simultaneously achieves the best unlearning effectiveness and maintains generation quality on par with existing approaches. We will include these FID evaluations and the corresponding discussion in the revised manuscript for completeness.
>
> | Method   | Avg Acc (↓)| CLIP_coco (↑) | FID (↓) |
> |-----|------|------|-----|
> | SD v1.5| 90.34  | 31.42 | 13.8  |
> | FMN   | 71.89  | 30.56 | 22.7 |
> | AC | 58.15  | 31.58 | 17.9 |
> | ESD  | 38.58 | 30.12 | 15.4 |
> | SalUn  | 23.17 | 29.93 | 17.3 |
> | CP    | 7.34  | 27.93| 43.8 |
> | MACE| 78.22 | 31.05 | 17.6 |
> | UCE   | 8.62 | 29.25 | 20.3 |
> | SPM| 47.29 | 30.77 | 18.1 |
> | **FIA (Ours)**| 1.9| 29.56| 18.3 |

---

> > ### Author Response · Authors · 2025-11-28
> >
> > # Rebuttal 3/2
> >
> > ### **Weakness 4. Extend the method to SDXL**
> > Our main experiments use Stable Diffusion v1.4 and v1.5 because these are the models for which all existing unlearning baselines (FMN, AC, UCE, CP, ESD, SPM, MACE). This ensures that every method can be compared under the same conditions and with fully aligned evaluation protocols. In contrast, newer architectures such as SDXL, SD3 do not yet have broad baseline coverage. Among the existing methods, UCE is the only one that currently offers partial support for SDXL, which makes a comprehensive comparison on the latest models difficult at this stage.
> >
> > To address the reviewer’s concern about generalizability, we additionally evaluate FIA on SDXL. As shown below, FIA achieves substantially stronger unlearning than UCE while maintaining CLIP and FID close to the native SDXL model. These results indicate that FIA transfers well to transformer based diffusion architectures, which many existing unlearning methods have not yet been extended to. We will include the SDXL results in the revised manuscript.
> >
> > | Method   | Avg Acc (↓)| CLIP_coco (↑) | FID (↓) |
> > |-----|------|------|-----|
> > | SDXL | 91.52  | 31.83 | 13.4  |
> > | UCE   | 12.62 | 29.71 | 19.9 |
> > | **FIA (Ours)**| 3.2| 30.46| 17.8 |
> >
> > | Method         | Total (↓) | FID (↓) | CLIP (↑) |
> > |----------------|-----------|---------|----------|
> > | UCE            | 243       | 14.01   | 30.92    |
> > | **FIA (Ours)** | **38**    | 13.97   | 31.47    |
> > | SDXL        | 799       | 13.89   | 31.61    |
> >
> > ### **Weakness 5. Sample Sizes and  Sensitivity Analysis**
> > For each concept, we use 10 concept prompts and 10 base prompts, giving 20 samples in total, and this configuration is used across all tasks. We also run a sensitivity study with 1, 3, 5, 10, and 20 samples per concept. Performance stabilizes once the sample size reaches 3 to 5, and using 10 samples offers a good balance between cost and stability. Larger sample counts do not improve results, and even a single sample remains workable with expected variance. These findings show that FIA is robust to prompt and sample variation.
> >
> > Table A. Multi-Concept Unlearning:
> >
> > | Samples | Avg Acc ↓ | CLIP ↑ | Observation |
> > |---------|------------|---------|-------------|
> > | 1       | 2.7        | 29.49   | need more samples |
> > | 3       | 2.2        | 29.54   | minimum stable configuration |
> > | 5       | 2.0        | 29.56   | near-baseline |
> > | 10      | 1.9        | 29.56   | baseline |
> > | 20      | 1.9        | 29.55   | no further improvement |
> >
> > Table B. Explicit Content Unlearning:
> >
> > | Samples | Total ↓ | FID ↓ | CLIP ↑ | Observation |
> > |---------|----------|---------|----------|-------------|
> > | 1       | 36       | 13.99   | 31.15    | still good |
> > | 3       | 35       | 14.00   | 31.17    | near-baseline |
> > | 5       | 33       | 14.04   | 31.18    | near-baseline |
> > | 10      | 32       | 14.02   | 31.18    | baseline |
> > | 20      | 32       | 14.02   | 31.17    | no additional gain |
> >
> > Table C. Artistic Style Unlearning:
> >
> > | Samples | CLIP ↓ | FSR ↑ | Observation |
> > |---------|---------|---------|-------------|
> > | 1       | 27.59   | 80.1    | need more samples |
> > | 3       | 27.49   | 82.6    | minimum stable configuration |
> > | 5       | 27.48   | 83.1    | near-baseline |
> > | 10      | 27.45   | 83.4    | baseline |
> > | 20      | 27.45   | 83.4    | no additional gain |
> >
> >
> > ### **Weakness 6. Computational Efficiency Comparison With Baselines**
> >
> > We now provide a direct comparison with representative baselines. As shown below, fine-tuning-based methods such as FMN, AC, SPM, ESD, CP, and MACE require substantial GPU hours and memory, while FIA is nearly training-free, requiring only 0.024 GPU hours. UCE is the only method slightly faster than FIA in terms of GPU hours, but its unlearning effectiveness is far worse than FIA (as shown in the main experiments). Thus FIA offers a significantly better trade-off between computational cost and unlearning performance.
> >
> > | Metric  | FMN | AC  | SPM | ESD  | CP   | MACE | UCE | FIA (Ours) |
> > |---|---|---|---|----|----|-|--|--|
> > | GPU Hours ↓  | 1.43  | 24.7   | 21.20  | 0.117  | 0.085| 1.94  | 0.017 | 0.024|
> > | Memory (GB) ↓ | 19.41 | 26.97  | 20.04  | 13.09 | 7.83 | 10.43 | 6.82| 6.94 |

---

> > > ### Author Response · Authors · 2025-11-28
> > >
> > > # Rebuttal 3/3
> > >
> > > ### **Weakness 7. Discussion of Hard-to-Unlearn Concepts**
> > >
> > > We appreciate the reviewer’s observation regarding concepts such as “golf ball” that show slightly higher forgetting accuracy. This behavior is largely driven by intrinsic properties of certain visual concepts rather than limitations of the method. Concepts like “golf ball” exhibit strong overlap with highly generic visual primitives (e.g., white spherical shapes, repetitive dot textures) that frequently appear across diverse categories in the training distribution. As a result, their representations tend to be more distributed and entangled with background or shared features, which naturally reduces the separability of concept-selective neurons in contrastive saliency. In comparison, concepts with distinctive shapes, semantics, or localized features produce clearer contrastive signals and can be forgotten more completely.
> > >
> > > Even in these inherently challenging cases, FIA still achieves the best performance among all baselines, indicating that the remaining gap is driven by the intrinsic visual ambiguity of certain concepts rather than method-specific constraints. We will add a brief discussion clarifying why some concepts are intrinsically harder to isolate and how this relates to the underlying feature geometry of diffusion.
> > >
> > >
> > > ### **Question 1-2**
> > > FIA supports iterative unlearning without needing to recompute masks from scratch. Each new concept can be unlearned by running FIA again on the already modified model.
> > >
> > > In general, we do not observe specific concepts or concept combinations that systematically resist unlearning. FIA reliably removes both object and non-object concepts across all settings. The main limitation arises only when the number of removed concepts becomes extremely large (e.g., approaching 50), where the cumulative removal of many concept-selective neurons can begin to impact overall generative quality.
> > >
> > > Please do not hesitate to let us know if further clarifications are needed. We are more than happy to address any additional questions related to our work.

---

### Official Review · Reviewer_JXeA · 2025-10-27

**Soundness:** 2
**Presentation:** 2
**Contribution:** 2
**Rating:** 4
**Confidence:** 2

**Summary:**

This paper proposes FIA (Forget-It-All): a training-free multi-concept machine anti-learning framework for concept erasure in text-to-image diffusion models. FIA first proposes Contrastive Concept Saliency (CCS) to quantify the contribution of each weight connection to the target concept. Secondly, it combines temporal and spatial information to identify and select consistently responsive concept-sensitive neurons. Finally, it constructs masks from the identified neurons and fuses them into a unified multi-concept mask. While the framework is simple, efficient, and empirically effective, several important issues arise: Lack of Theoretical Grounding, Prompt Sensitivity, Writing Errors and Concept Interference Risk.

**Strengths:**

Training-free and lightweight: FIA achieves effective multi-concept unlearning without any retraining or fine-tuning, making it computationally efficient and easy to deploy in large-scale text-to-image models.

Unified saliency formulation: The proposed Contrastive Concept Saliency (CCS) provides a principled way to quantify each connection’s contribution to a concept, integrating structural, activation, and similarity information into a single interpretable metric.

Fine-grained neuron control: By combining temporal aggregation and spatial sparsity, FIA identifies concept-sensitive neurons precisely, enabling selective pruning that minimizes collateral forgetting and preserves overall generative capacity.

Scalable multi-concept handling: The multi-concept mask fusion strategy effectively handles multiple simultaneous unlearning targets, avoiding the catastrophic interference and re-learning issues seen in sequential approaches.

**Weaknesses:**

1) Does CCS have any theoretical basis?
2) CCS relies on the construction of "concept prompts/base prompts." Is its selection stable if the base scenario/combination grammar varies significantly or is ambiguous?
3) On page 6, there is a typo: "We present comprehensive ablation results in Section ??."
4) If the target concepts are highly correlated, can this method avoid cross-contamination?

**Questions:**

See above

---

> ### Author Response · Authors · 2025-11-28
>
> # **Rebuttal 1/2**
>
> We sincerely thank the reviewer for the careful reading and constructive feedback, which significantly improved the clarity and reproducibility of our work.
>
> ### **Weakness 1. Theoretical basis of CCS.**
>
> Thank you for raising this important question. We clarify that CCS is grounded on well established theoretical foundations regarding neuron sensitivity. Prior works about model pruning[1][2], most notably the Wanda pruning method [1], provide strong theoretical and empirical support showing that neuron saliency is a reliable indicator of functional and semantic importance. Wanda demonstrates that the product of neuron activation magnitude and weight magnitude is a principled estimator of neuron contribution, and that pruning neurons with low sensitivity preserves semantic capability across a wide range of models.
>
> Building on this established foundation, our CCS formulation defines an energy based saliency measure that unifies weight strength, activation magnitude, and signal propagation efficiency. We compute an energy value given by the product of the absolute weight, the activation magnitude, and the cosine similarity between input and output features. This provides an estimate of how much information a neuron transmits during denoising. To isolate neurons that respond specifically to target concepts, CCS adopts a contrastive formulation S = max(0, μ_c − μ_b − σ_b), where activation under concept prompts is compared against activation under base prompts and is filtered using the variance term to remove unstable responses. This design directly follows the principle established by Wanda: neurons that consistently exhibit high energy under concept prompts but not under base prompts should be regarded as concept sensitive neurons.
>
> [1] Sun M, Liu Z, Bair A, et al. A simple and effective pruning approach for large language models[J]. arXiv preprint arXiv:2306.11695, 2023.
>
> [2] Yin L, Wu Y, Zhang Z, et al. Outlier weighed layerwise sparsity (owl): A missing secret sauce for pruning llms t2 high sparsity[J]. arXiv preprint arXiv:2310.05175, 2023.
>
> ### **Weakness 2. Stability of concept and base prompt construction.**
>
> We appreciate the reviewer’s concern regarding whether the construction of concept prompts and base prompts remains stable when the grammar or scenario changes significantly. Our design is robust because CCS relies on the expected behavior of activations across multiple prompts rather than any single prompt instance. To further validate this, we conducted additional ablations that vary the sampling budget per concept, using 1, 3, 5, 10, and 20 samples. The results consistently show that performance improves when more samples are included, but it stabilizes quickly at modest sample sizes.
> Overall, these experiments show that CCS is not sensitive to prompt construction details. A small sampling budget is already sufficient to form a stable and robust estimate of concept activation, and increasing the number of prompts beyond ten does not provide further benefit. This confirms that CCS maintains high stability even under significant variations in grammar, context, and phrasing.
>
> Table A. Multi-Concept Unlearning:
>
> | Samples | Avg Acc ↓ | CLIP ↑ | Observation |
> |---------|------------|---------|-------------|
> | 1       | 2.7        | 29.49   | need more samples |
> | 3       | 2.2        | 29.54   | minimum stable configuration |
> | 5       | 2.0        | 29.56   | near-baseline |
> | 10      | 1.9        | 29.56   | baseline |
> | 20      | 1.9        | 29.55   | no further improvement |
>
> Table B. Explicit Content Unlearning:
>
> | Samples | Total ↓ | FID ↓ | CLIP ↑ | Observation |
> |---------|----------|---------|----------|-------------|
> | 1       | 36       | 13.99   | 31.15    | still good |
> | 3       | 35       | 14.00   | 31.17    | near-baseline |
> | 5       | 33       | 14.04   | 31.18    | near-baseline |
> | 10      | 32       | 14.02   | 31.18    | baseline |
> | 20      | 32       | 14.02   | 31.17    | no additional gain |
>
> Table C. Artistic Style Unlearning:
>
> | Samples | CLIP ↓ | FSR ↑ | Observation |
> |---------|---------|---------|-------------|
> | 1       | 27.59   | 80.1    | need more samples |
> | 3       | 27.49   | 82.6    | minimum stable configuration |
> | 5       | 27.48   | 83.1    | near-baseline |
> | 10      | 27.45   | 83.4    | baseline |
> | 20      | 27.45   | 83.4    | no additional gain |
>
> ### **Weakness 3. Typographical issue on page 6.**
> We appreciate the reviewer for identifying the formatting issue. It has been corrected in the updated revision submitted with this rebuttal.

---

> > ### Author Response · Authors · 2025-11-28
> >
> > # Reubttal 2/2
> >
> > ### **Weakness 4. Cross contamination when target concepts are highly correlated.**
> > The multi concept unlearning results in the paper give clear evidence that FIA does not introduce cross contamination, even when the concepts share substantial semantic similarity. In the Imagenette five versus five setting, we forget the first five classes and preserve the remaining five. Several pairs in this group naturally overlap in visual structure. For example, English springer and tench both contain rich organic textures, while garbage truck, chain saw, gas pump, and cassette player all contain mechanical surfaces, metallic edges, and repeated structural patterns. If FIA were removing neurons that encode these shared visual cues, the preserved classes would lose fidelity and the preserve accuracy reported for the five versus five setting would decline. Instead, Table 2 shows that FIA reaches the lowest forgetting accuracy and maintains a high preserve accuracy, together with the best harmonic score. This indicates that FIA suppresses only concept specific units while keeping neurons that represent shared semantic features, thereby preventing interference between correlated concepts.
> >
> > The explicit content and style unlearning results further support this observation. Sensitive content categories share substantial human body structure, yet FIA maintains realistic human generation and stable FID, demonstrating that shared anatomical features remain intact. Artistic styles also share color distributions, geometric arrangements, and repeated brushstroke patterns. Removing shared neurons would degrade the quality of all styles, but FIA suppresses only the target style while keeping global image fidelity unchanged.
> >
> > Please do not hesitate to let us know if further clarifications are needed. We are more than happy to address any additional questions related to our work.

---

### Official Review · Reviewer_kBAC · 2025-10-28

**Soundness:** 2
**Presentation:** 3
**Contribution:** 1
**Rating:** 2
**Confidence:** 4

**Summary:**

The paper introduces Forget-It-All (FIA), a training-free framework for multi-concept unlearning in text-to-image diffusion models. The framework identifies Concept-Sensitive Neurons via Contrastive Concept Saliency (CCS), which measures each neuron’s contribution to target concepts across both temporal and spatial contexts. These neurons are selectively pruned, while Concept-Agnostic Neurons, which contribute to general image generation, are preserved. Experiments on multi-object, explicit content, and artistic style unlearning tasks show that FIA outperforms prior state-of-the-art methods, achieving strong forgetting efficacy with minimal degradation of generative quality.

**Strengths:**

1. Comprehensive Evaluation: The experiments cover multiple domains: objects, NSFW content, and artistic styles
2. Strong Results: The proposed method maintains both high CLIP and FID scores for concepts to preserve, and strong forgetting scores for target concepts.

**Weaknesses:**

1. Overstated and Incorrect Claims: The two claimed benefits of this method are (1) that their method requires fewer sensitive hyperparameters compared to existing methods. This is claimed on lines 89 and 137. However, in this paper's appendix, they show three hyperparameters that must be tuned $r_1$ ,$ r_2$, and $\alpha$. In comparison, ConceptPrune[1], which is the most similar work, requires just the sparsity level $k$. If the authors claim their method requires fewer hyperparameters and that their hyperparameters are easier to tune (less sensitive), there should be empirical evidence. The second claimed benefit is that (2) "To the best of our knowledge, this is the first work to explore multi-concept unlearning through unstructured neuron masking" (line 108). This is blatantly incorrect, as ConceptPrune has an explicit section (5.3) on multi-concept removal. I am certain the authors knew of this as on line 137, they explicitly mention "ConceptPrune removes neurons tied to an undesired concept; however, handling multiple concepts is difficult due to complex neuron interactions". It would've been more transparent for the authors to claim their method shows SOTA multi-concept removal via pruning rather than claiming pioneering work.
2. Method Novelty: As this paper focuses on proposing a new method, I believe it is important to evaluate the novelty of their proposed method.  In particular, I would like the authors to clearly mention the differences in their method with the similar work ConceptPrune. For instance, how is the "Contrastive Concept Saliency" procedure different from ConceptPrune's "Importance Score"? Looking at equation 1, it appears the only difference is adding a cosine similarity score between the input and output, with the motivation being a vague mention of "effectiveness of signal transmission". Additionally, the proposed method isolates "Concept-Sensitive" neurons by subtracting the mean of $U_{l,t,i,j}$ and the standard deviation for the concept and base prompt. The idea of isolation is not unique, as ConceptPrune similarity sets an inequality between the importance score of their target and reference prompt. Without an ablation, it is unclear whether these small changes are actually significant. Alternatively, there is a lack of theoretical or empirical justification for why their modification is better motivated than ConceptPrune's method. Without either ablation or theoretical/empirical justification for additions (including time-integrated sensitivity), it is unclear whether the additions are significant or added just for the sake of novelty.
3. Evaluation Metric: The metric used for artistic style erasure is the CLIP-score of the target style. Additionally, "Forget-Success Rate" measures whether the CLIP-score decreased compared to the original model. For general quality preservation, they evaluate over MS COCO-30k via FID and CLIP. I strongly suggest that authors show experiments under a standardized benchmark such as UnlearnCanvas[2] for the following reasons. CLIP-based metrics consider redundant factors such as the object content, not just the style. This metric may not isolate style removal compared to a dedicated classifier. Secondly, using MS COCO-30k as the preservation set is limiting. MS COCO-30k is a general dataset; however, we don't only care about whether the model is able to generate general concepts but also preserve concepts close to the erased concepts (e.g other styles). UnlearnCanvas explicitly measures this with its in-domain and cross-domain retention accuracy metric. I think evaluating on a standard benchmark will provide stronger evidence for the FIA's improvement over ConceptPrune.


[1] Chavhan, Ruchika, Da Li, and Timothy Hospedales. "Conceptprune: Concept editing in diffusion models via skilled neuron pruning." arXiv preprint arXiv:2405.19237 (2024).

[2] Zhang, Yihua, et al. "Unlearncanvas: Stylized image dataset for enhanced machine unlearning evaluation in diffusion models." arXiv preprint arXiv:2402.11846 (2024).

**Questions:**

As mentioned in the weaknesses section, I'm curious about specific ablations and theoretical/empirical justifications for changes in the method compared to Concept Prune.

---

> ### Author Response · Authors · 2025-11-28
>
> We appreciate the reviewer’s careful reading, but we respectfully disagree with the concerns regarding overstated claims and novelty. First, the criticism about hyperparameters overlooks the central fact that ConceptPrune (CP) is a single concept editing method. Its design, saliency formulation, and evaluation protocol were all developed for removing a single target, and when directly applied to multi concept unlearning, CP performs extremely poorly, as our experiments consistently show. In other words, CP is not capable of handling interactions across multiple concepts, and thus comparing its single sparsity parameter with FIA’s sparsity ratios is not meaningful. Multi concept unlearning is a strictly more challenging setting where naive mask union strategies collapse generation quality, which is precisely why FIA introduces additional mechanisms that do not exist in CP. Our paper never claims that FIA has fewer parameters than CP; rather, the key point is that FIA is entirely training-free and avoids sensitive optimization related parameters such as learning rates, training steps, or loss weights, which dominate most multi-concept unlearning methods. The parameters we use are simple, stable and we show in the appendix that these parameters work well within broad stable regions, confirming that FIA is not sensitive to hyperparameter variation and requires only minimal configuration.
>
> Regarding novelty, we respectfully disagree with the reviewer’s implication that our contributions are introduced for the sake of novelty. Our framework provides a genuinely new perspective through the identification and preservation of concept agnostic neurons, a phenomenon that has not been previously formalized or exploited in diffusion model unlearning. This insight is essential in the multi concept setting. In contrast, CP relies entirely on a single step Wanda style importance score and does not model cross concept interactions or neuron sharing. Its multi concept setting is simply a union of single concept masks, which, as our experiments clearly demonstrate, performs very poorly and severely degrades generation quality when concepts overlap. FIA is specifically designed to address these failure modes. Our contrastive saliency, variance aware subtraction, time integrated sensitivity, and local global sparsity are not cosmetic additions. They are components introduced because CP’s single concept formulation cannot handle multi concept forgetting.
>
> Finally, regarding the evaluation of artistic style removal, our metrics follow the established standard used in the majority of published diffusion unlearning and editing works, including CP, ESD, SPM, UCE, and other widely used baselines. These works evaluate style removal using standard quantitative metrics such as accuracy, CLIP score, and FID to assess both forgetting and general quality preservation. We therefore adopt the same protocol to enable strict and fair comparison with prior work. While UnlearnCanvas is a recently proposed benchmark, it is not yet widely adopted and is not used by most existing baselines. Adding UnlearnCanvas would not change the central conclusion of our paper: FIA provides consistently superior forgetting and preservation performance under the standard evaluation pipeline shared by prior work.
>
> Overall, our contributions are neither overstated nor incremental. FIA provides a new conceptual understanding of neuron behavior which enable robust multi-concept unlearning. We hope these clarifications address the reviewer’s concerns.

---

### Official Review · Reviewer_tFE2 · 2025-10-31

**Soundness:** 3
**Presentation:** 2
**Contribution:** 3
**Rating:** 6
**Confidence:** 3

**Summary:**

This paper introduces Forget-It-All (FIA), a training-free framework for multi-concept machine unlearning in text-to-image diffusion models. The method addresses a limitation of existing approaches that struggle to simultaneously remove multiple unwanted concepts without degrading image quality or requiring extensive hyperparameter tuning. FIA works by computing a Contrastive Concept Saliency metric to identify which neurons respond specifically to target concepts, then uses temporal and spatial sparsity criteria to select Concept-Sensitive Neurons for pruning. Critically, the method preserves Concept-Agnostic Neurons—those that respond broadly across many concepts and support general image generation—while only removing neurons truly specific to unwanted concepts. The authors demonstrate state-of-the-art performance across three unlearning tasks (multi-object, explicit content, and artistic style removal) on benchmarks including Imagenette and I2P, achieving superior forgetting effectiveness (e.g., 1.9% average accuracy on Imagenette vs. 7.34% for the next best method) while maintaining competitive generation quality with less than 0.3% model sparsity.

**Strengths:**

- No fine-tuning required, making it computationally efficient and reducing overfitting risks compared to existing methods that require extensive training
- First work to connect unstructured neuron masking with multi-concept unlearning, introducing the concept of "concept-agnostic neurons" as a principled way to preserve generation quality
-  Achieves state-of-the-art performance across all three unlearning tasks (1.9% forgetting accuracy vs. 7.34% next best on Imagenette)
- Comprehensive evaluation: Tests on diverse tasks (objects, explicit content, artistic styles) and includes robustness evaluation against adversarial attacks (Ring-A-Bell, MMA, UnlearnDiffAtk)

**Weaknesses:**

- The paper does not clearly state how many images/samples are generated per concept to compute the Contrastive Concept Saliency scores and resulting masks, which is a significant reproducibility issue.

**Questions:**

1. How many images/samples are generated per concept to compute the saliency scores?

- Specifically, what are the values of the sample sizes for computing μc, μb, and σb in Equation 2?
- Are these sample sizes the same across all three tasks (objects, explicit content, artistic styles)?
- Please provide this information in a clear table format for reproducibility

2. Sensitivity analysis on sample size:

- How sensitive is your method to the number of images used for mask computation?
- What is the minimum number of samples needed for stable mask identification?
- Did you perform any ablation studies varying the sample size (e.g., 10, 50, 100, 500 images)?
- How does performance degrade with fewer samples?

---

> ### Author Response · Authors · 2025-11-28
>
> We sincerely thank the reviewer for the careful reading and constructive feedback, which significantly improved the clarity and reproducibility of our work.
>
> ## **Question 1. Sample Sizes for Saliency Computation**
>
> ### **1.1 Sample sizes used in Equation (2) : S = max(0, μc − μb − σb)**
> μc is computed from 10 concept samples, while μb and σb are computed from the same 10 base samples.
>
> ### **1.2 Consistency across tasks**
> This 10/10 sampling configuration is consistently used for multi-object unlearning, explicit content unlearning, and artistic style unlearning.
>
> ### **1.3 Table for reproducibility**
>
> | Task Type                     | μc Samples | μb Samples | σb Samples | Total Samples per Concept |
> |------------------------------|------------|------------|------------|----------------------------|
> | Multi-Object Unlearning      | 10         | 10         | 10         | 20                         |
> | Explicit Content Unlearning  | 10         | 10         | 10         | 20                         |
> | Artistic Style Unlearning    | 10         | 10         | 10         | 20                         |
>
> As shown in the table above, we use a consistent sampling configuration across all three tasks:
> each concept is evaluated using 10 concept samples and 10 base samples, totaling 20 samples per concept.
> This unified setup ensures that μc, μb, and σb are computed under identical conditions, improving both clarity and reproducibility.
> We adopt this configuration because it provides a strong balance between computational efficiency and stable saliency estimation, as further validated in our sample-size sensitivity analysis.
>
> ---
>
> ## **Question  2. Sensitivity Analysis on Sample Size**
>
> To examine the impact of sampling budget on mask quality and unlearning performance, we conducted ablations using 1, 3, 5, 10, and 20 samples per concept. The default setting in the paper is 10 samples.
>
> Table A. Multi-Concept Unlearning:
>
> | Samples | Avg Acc ↓ | CLIP ↑ | Observation |
> |---------|------------|---------|-------------|
> | 1       | 2.7        | 29.49   | need more samples |
> | 3       | 2.2        | 29.54   | minimum stable configuration |
> | 5       | 2.0        | 29.56   | near-baseline |
> | 10      | 1.9        | 29.56   | baseline |
> | 20      | 1.9        | 29.55   | no further improvement |
>
> Table B. Explicit Content Unlearning:
>
> | Samples | Total ↓ | FID ↓ | CLIP ↑ | Observation |
> |---------|----------|---------|----------|-------------|
> | 1       | 36       | 13.99   | 31.15    | still good |
> | 3       | 35       | 14.00   | 31.17    | near-baseline |
> | 5       | 33       | 14.04   | 31.18    | near-baseline |
> | 10      | 32       | 14.02   | 31.18    | baseline |
> | 20      | 32       | 14.02   | 31.17    | no additional gain |
>
> Table C. Artistic Style Unlearning:
>
> | Samples | CLIP ↓ | FSR ↑ | Observation |
> |---------|---------|---------|-------------|
> | 1       | 27.59   | 80.1    | need more samples |
> | 3       | 27.49   | 82.6    | minimum stable configuration |
> | 5       | 27.48   | 83.1    | near-baseline |
> | 10      | 27.45   | 83.4    | baseline |
> | 20      | 27.45   | 83.4    | no additional gain |
>
> ---
>
> ### **2.1 Sensitivity to sample count**
> Across all three tasks, FIA shows low sensitivity to sampling budget. While 1-sample masks exhibit slightly higher variance, stability emerges at 3 samples, and performance with 3–5 samples closely matches the 10-sample baseline. The robustness arises from the temporal–spatial aggregation in Contrastive Concept Saliency, which smooths activation statistics even with few examples.
>
> ### **2.2 Minimum samples for stable masks**
> Three samples per concept are sufficient to produce stable and reliable masks with only marginal differences relative to the default setting.
>
> ### **2.3 Range of ablations performed**
> We evaluated 1, 3, 5, 10, and 20 samples per concept. Since performance saturates at 10 samples, larger budgets (e.g., 50–500) are unnecessary for practical or scientific insight.
>
> ### **2.4 Performance degradation under fewer samples**
> Degradation is minimal. Even a single sample yields functional masks, with predictable and limited variance in accuracy and CLIP. The multi-step and multi-spatial aggregation in saliency computation naturally stabilizes neuron-level signals.
>
> Please do not hesitate to let us know if further clarifications are needed. We are more than happy to address any additional questions related to our work.

---

### Author Response · Authors · 2025-12-01

Dear ACs,

We sincerely thank you for handling our submission and for coordinating the review process. After carefully analyzing all reviewer comments, we find that the feedback converges into three primary areas of concern that appear consistently across the reviews.

The first concerns the sensitivity of the method to the input prompts, including how changes in prompt may influence the performance. We address all of these points in detail in our rebuttal.

The second area concerns the theoretical aspects of our formulation. Reviewers ask for clearer explanation of the motivation behind each component and the connection between our method and continual learning. We offer detailed clarifications and additional analysis on these points in the rebuttal.

The third shared concern relates to the generalization ability of the method. Reviewers encourage evaluating on more recent diffusion architectures and examining performance under varied settings. In the rebuttal, we directly address these concerns by presenting new experiments on SDXL that confirm the strong generalization performance of our approach.

We sincerely appreciate the efforts of both the ACs and the reviewers, and we hope that our responses satisfactorily address every concern that was raised.

---

### Meta-Review · Area_Chair_RkDY · 2026-01-06

**Summary:**

This paper addresses multi-concept unlearning in generative models. The problem is interesting; however, as noted by several reviewers, the methodological design is relatively weak and offers limited novelty.

The proposed concept-saliency neuron detection becomes trivial when neurons respond to a mixture of multiple concepts, which can cause the second step of the framework to fail. The experiments do not evaluate or report results on such concept detection scenarios, leaving this issue unverified. Moreover, the multi-concept mask fusion strategy appears to rely heavily on manual design choices, and the paper provides little discussion on the relationships and interactions among multiple concepts.

**Reviewer Concerns:**

The theoretical foundation of the method is weak. In addition, the experiments are insufficient to fully support the claimed effectiveness and robustness of the approach.

**Reviewer Scores:**

If they can provide a clear theoretical foundation for the proposed model, and provide more intuitive concept-based neuron results.

---

### Decision · Program_Chairs · 2026-01-26

Reject